# No imputation without representation

## Abstract

By filling in missing values in datasets, imputation allows these datasets to be used with algorithms that cannot handle missing values by themselves. However, missing values may in principle contribute useful information that is lost through imputation. The missing-indicator approach can be used in combination with imputation to instead represent this information as a part of the dataset. There are several theoretical considerations why missing-indicators may or may not be beneficial, but there has not been any large-scale practical experiment on real-life datasets to test this question for machine learning predictions. We perform this experiment for three imputation strategies and a range of different classification algorithms, on the basis of twenty real-life datasets. We find that on these datasets, missing-indicators generally increase classification performance. In addition, we find no evidence for most algorithms that nearest neighbour and iterative imputation lead to better performance than simple mean/mode imputation. Therefore, we recommend the use of missing-indicators with mean/mode imputation as a safe default, with the caveat that for decision trees, pruning is necessary to prevent overfitting. In a follow-up experiment, we determine attribute-specific missingness thresholds for each classifier above which missing-indicators are more likely than not to increase classification performance, and observe that these thresholds are much lower for categorical than for numerical attributes. Finally, we argue that mean imputation of numerical attributes may preserve some of the information from missing values, and we show that in the absence of missing-indicators, it can similarly be useful to apply mean imputation to one-hot encoded categorical attributes instead of mode imputation.

## 1 Introduction

Missing values are a frequent issue in real-life datasets, and the subject of a large body of ongoing research. Some implementations of machine learning algorithms can handle missing values natively, requiring no further action by practitioners. But whenever this is not the case, a common general strategy is to replace the missing value with an estimated value: imputation. An advantage of imputation is that we obtain a complete dataset, to which we can apply any and all algorithms that make no special provision for missing values. However, missing values may be informative, and a disadvantage of imputation is that it removes this information.

The missing-indicator approach (Cohen, 1968) is a well-established way to represent and thereby preserve the information encoded by missing values. For every original attribute, this approach adds a new binary 'indicator' or 'dummy' attribute that takes a value of 1 if the value for the original attribute is missing, and 0 if not.[1] The missing-indicator approach is often presented as an alternative to imputation, but since it does not resolve the missing values in the original attributes, it can only be used in addition to, not instead of imputation.

Both imputation and the missing-indicator approach originate in the statistical literature. While imputation strategies have been the subject of a rich body of research, the missing-indicator approach has not received a large amount of attention, and is often dismissed or disregarded in overviews of approaches towards missing

---

[1] Some authors use the opposite convention, letting the indicator express non-missingness.

values. In particular, it is an open question whether missing-indicators should be used for predictive tasks in machine learning (Sperrin et al., 2020). On the one hand, the addition of missing-indicators results in a more complete, higher-dimensional representation of the data. On the other hand, their omission can be seen as a form of dimensionality reduction, which may increase the efficiency and effectiveness of a dataset by eliminating redundancy.

To determine whether this trade-off is useful, a key question is to which extent missing values in a given dataset are informative, and to which extent their distribution is predictable. In the latter case, the phrase "missing at random" (MAR) (Rubin, 1976) is used to indicate that the distribution of missing values is dependent on the known values, while the stricter phrase "missing completely at random" (MCAR) denotes values that are distributed truly randomly. In contrast, informative missing values are often denoted as "missing not at random" (MNAR). For real-life datasets, unless we have specific knowledge about the process responsible for the missing values, we have to assume some degree of informativeness in principle.[2] However, Schafer (1997) has argued that in practice, the attributes of a dataset can be sufficiently redundant that one can get away with assuming its missing values are MAR. But even if this is so, imputation may not always perform optimally, in which case missing-indicators may still prove useful.

A more subtle point is that even when missing values are informative, the information they encode need not be lost completely through imputation. This is particularly evident in the case of numerically encoded binary attributes, where imputation can represent missing values as a third, intermediary value. More generally, Le Morvan et al. (2021) have observed that almost all deterministic imputation functions map records with missing values to distinct manifolds in the attribute space that can in principle be identified by sufficiently powerful algorithms. Nevertheless, including missing-indicators can still potentially make this learning task easier.

In light of these conflicting theoretical arguments, the usefulness of missing-indicators for real-life machine learning problems is an interesting empirical question. However, previous experiments in this direction have been limited in scope and number. These limitations include the use of only one or a handful of datasets, the use of datasets from which values have been removed artificially, at random (corresponding to the MCAR setting), and not comparing the same imputation strategies with and without missing-indicators.

The purpose of the present paper is straightforward. On the basis of twenty real-life classification problems with naturally occurring missing values, we evaluate the effect of missing-indicators on the performance of a range of popular classification algorithms, paired with three common types of imputation.

In Section 2, we provide a brief overview of the existing literature on missing-indicators, including previous experimental evaluations. In Section 3, we describe our experimental setup. We report our results in Section 4 and conclude in Section 5.

## 2 Background

We start with a brief discussion of the origins and reception of the missing-indicator approach, as well as previous experimental evaluations of the use of missing-indicators in prediction tasks.

### 2.1 Origins and reception

The missing-indicator approach originates in the statistical literature on linear regression, and dates back to at least Cohen (1968). Cohen pointed out that values in real-life datasets are typically not missing completely at random, and that the distribution of missing values may in particular depend on the values of the attribute that is to be predicted. He proposed that each attribute could be said to have two 'aspects', its value, and whether that value is present to begin with, which should be encoded with a pair of variables. For missing attribute values, the first of these variables was to be filled in with the mean of the known values, although other applications might call for different values. Cohen's proposal was subsequently expanded in Cohen

---

[2]This is acknowledged by authors working under the assumption of MAR, e.g. "When data are missing for reasons beyond the investigator's control, one can never be certain whether MAR holds. The MAR hypothesis in such datasets cannot be formally tested unless the missing values, or at least a sample of them, are available from an external source." (Schafer, 1997)

& Cohen (1975), but received only limited recognition in the following years (Kim & Curry, 1977; Stumpf, 1978; Chow, 1979; Hutcheson & Prather, 1981; Anderson et al., 1983; Orme & Reis, 1991).

Cohen's proposal was subjected to a formal analysis by Jones (1996), who showed that, if one assumes that missing values are MAR, and the true linear regression model does not contain any terms related to missingness, it produces biased estimates of the regression coefficients (unless the sample covariance between independent variables is zero). However, these assumptions run directly counter to the position set out by Cohen & Cohen (1975) that a priori, the missingness of each attribute is a possible explanatory factor, that it is safer not to assume that missing values are distributed randomly, and that the usefulness of missing-indicators is ultimately an empirical question.

Allison (2001), motivated by Jones (1996) and working under the general assumption of MAR, dismissed missing-indicators as "clearly unacceptable", before conceding that they in fact produce optimal estimates when the missing value is not just missing, but cannot exist, such as the marital quality of an unmarried couple. However, this semantic distinction may not always be clear-cut in practice, and the more pertinent question may be whether missing values are informative. Allison (2010) later acknowledged that missing-indicators may lead to better predictions and their use for that purpose was acceptable. Missing-indicators have also been dismissed by Pigott (2001); Schafer & Graham (2002); Graham (2009); Aste et al. (2015), and are frequently omitted in overviews of missing data strategies (Schafer, 1997; Enders, 2010; Eirola, 2014; García et al., 2015; Das et al., 2018).

## 2.2 Previous experiments

Only a handful of experimental comparisons of missing data approaches have included the missing-indicator approach, and these have been limited in scope. Vamplew & Adams (1992) and Ng & Yusoff (2011) only use a single dataset with randomly removed values, and base their evaluation on the performance of a single algorithm (respectively a neural network and linear regression). Pereira Barata et al. (2019) use three classification algorithms and 22 datasets, but again with randomly removed values, explicitly assuming a MCAR context. They conclude that imputation outperforms missing-indicators, but the comparison is not like-for-like, since it involves several forms of imputation but only combines indicator attributes with zero imputation. Van der Heijden et al. (2006) compare missing-indicators with zero imputation against several other forms of imputation without missing-indicators on one real dataset, for logistic regression. However, it appears that they do not use a test set, and only evaluate the resulting models on the training set.

Ding & Simonoff (2010) conduct a more extensive investigation, using insights from a series of Monte Carlo simulations to systematically remove values from 36 datasets to simulate different forms of missingness. They use these datasets to compare zero imputation[3] with indicator attributes against mean/mode imputation without, as well as a number of other missing data approaches, for logistic regression. In addition, the authors evaluate a related representation of missing values[4] on the same set of 36 datasets, and on one real-life dataset with missing values, for decision trees. They find that there is strong evidence that representing missing values is the best approach when they are informative; when this is not the case their results show no strong difference with respect to imputation.

The comparison by Grzymala-Busse & Hu (2000) is based on 10 datasets with naturally occurring missing values. However, the setting is purely categorical — all attributes are transformed into categorical attributes — the only form of imputation is mode imputation, and the missing value approaches are evaluated on the basis of the LERS classifier (Learning from Examples based on Rough Sets (Grzymala-Busse, 1988)).

Marlin (2008) compares zero imputation with missing-indicators (*augmentation with response indicators*) against several forms of imputation without, for logistic regression and neural networks, on the basis of an extensive series of simulations, one dataset with artificially removed values, and three real datasets. For the real datasets, there is no strong difference in performance between the different approaches.

---

[3]Presumably, Ding & Simonoff (2010) use one-hot encoding for categorical attributes, in which case zero imputation is equivalent to treating missing values as a separate category, but they do not state this explicitly.

[4]For categorical values, encoding missing values as a separate category, for numerical values, encoding missing values as an extremely large value that can always be split from the other values.

Most recently, Josse et al. (2020) and Le Morvan et al. (2021) respectively evaluate missing-indicators (*missingness mask*) for regression trees, Random Forest and XGBoost, and for multilayer perceptrons, on simulated regression datasets, and conclude that when missing values are informative, using missing-indicators clearly increases performance. This work has been continued by Perez-Lebel et al. (2022), who compare four different imputation techniques with and without missing-indicators on seven prediction tasks derived from four real medical datasets, and conclude that missing-indicators consistently improve performance for gradient boosted trees, ridge regression and logistic regression.

We point out that the Missingness in Attribute (MIA) proposal (Twala et al., 2008) for decision trees and decision tree ensembles can be understood as an implicit combination of missing-indicators with automatic imputation, and has also been shown to outperform imputation without missing-indicators in small-scale experimental studies (Josse et al., 2020; Perez-Lebel et al., 2022).

Finally, even experimental comparisons of missing data that do not feature the missing-indicator approach generally do not involve more than a handful of real-life datasets with naturally occurring missing values. We have only found Luengo et al. (2012a;b), who use 21 datasets from the UCI repository, but 12 of these are problematic.[5]

## 3 Experimental setup

To evaluate the effect of the missing-indicator approach on classification performance, we conduct a series of experiments, using the Python machine learning library *scikit-learn* (Pedregosa et al., 2011).

### 3.1 Questions

The aim of our experiments is to answer the following questions:

- Do missing-indicators increase performance, and does it matter which imputation strategy they are paired with?

- When do missing-indicators start to become useful in terms of missingness?

- Does using mean imputation instead of mode imputation allow for more information to be learned from missing categorical values?

### 3.2 Evaluation

We preprocess datasets by standardising numerical attributes and one-hot encoding categorical attributes (as required by the implementations in scikit-learn).

We measure classification performance by performing stratified five-fold cross-validation, repeating this for five different random states (which determine both the dataset splits and the initialisation of algorithms with a random component), and calculating the mean area under the receiver operator curve (AUROC). For multi-class datasets, we use the extension of AUROC defined by Hand & Till (2001).

To compare two alternatives A and B, we consider the *p*-value of a one-sided Wilcoxon signed-rank test (Wilcoxon, 1945) on the mean AUROC scores for our selection of datasets. When we compare A vs B, a

---

[5]The target column of the *echocardiogram* dataset ('alive-at-1') is supposed to denote whether a patient survived for at least one year, but it doesn't appear to agree with the columns from which it is derived, that denote how long a patient (has) survived and whether they were alive at the end of that period. The *audiology* dataset has a large number of small classes with complex labels and should perhaps be analysed with multi-label classification. In addition, it has ordinal attributes where the order of the values is not entirely clear, and three different values that potentially denote missingness ('?', 'unmeasured' and 'absent'), and it is not completely clear how they relate to each other. The *house-votes-84* dataset contains '?' values, but its documentation explicitly states that these values are not unknown, but indicate different forms of abstention. The *ozone* dataset is a time-series problem, while the task associated with the *sponge* and *water-treatment* datasets is clustering, with no obvious target for classification among their respective attributes. Finally, the *breast-cancer* (9), *cleveland* (7), *dermatology* (8), *lung-cancer* (5), *post-operative* (3) and *wisconsin* (16) datasets contain only very few missing values, and any performance difference between missing value approaches on these datasets may to a large extent be coincidental.

Table 1: Classification algorithms.

| Name | Description |
|------|-------------|
| NN-1 | Nearest neighbours (Fix & Hodges, 1951) with (Boscovich) 1-distance |
| NN-2 | Nearest neighbours with (Euclidean) 2-distance |
| NN-1-D | Nearest neighbours with 1-distance, distance-weighted (Dudani, 1976) |
| NN-2-D | Nearest neighbours with 2-distance, distance-weighted |
| SVM-L | Soft-margin Support Vector Machine (Cortes & Vapnik, 1995) with linear kernel |
| SVM-G | Soft-margin Support Vector Machine with Gaussian kernel |
| LR | Multinomial logistic regression (Cox, 1966) |
| MLP | Multilayer perceptron (Rosenblatt, 1961) with ReLu activation (Fukushima, 1969), Glorot initialisation (Glorot & Bengio, 2010) and Adam optimisation (Kingma & Ba, 2014) |
| CART | Classification and Regression Tree (Breiman et al., 1984) |
| RF | Random Forest (Breiman, 2001) |
| ERT | Extremely Randomised Trees (Geurts et al., 2006) |
| ABT | Ada-boosted trees (Freund & Schapire, 1995) with SAMME (stagewise additive modeling using a multi-class exponential loss function) (Zhu et al., 2009) |
| GBM | Gradient Boosting Machine (Friedman, 2001) |

score below 0.5 means that A increased performance on our selection of datasets; the lower the scores, the more confident we can be that this generalises to other similar datasets. Conversely, a score higher than 0.5 means that A decreased performance on our selection of datasets.

### 3.3 Imputation Strategies

We consider the following three imputation strategies:

- *Mean/mode imputation* replaces missing values of numerical and categorical attributes by, respectively, the mean and the mode of the non-missing values.

- *Nearest neighbour imputation* (Troyanskaya et al., 2001) replaces missing values of numerical and categorical attributes by, respectively, the mean and the mode of the 5 nearest non-missing values, with distance determined by the corresponding non-missing values for the other attributes.

- *Iterative imputation*, as implemented in scikit-learn, based on Van Buuren & Groothuis-Oudshoorn (2011), predicts missing values of one attribute on the basis of the other attribute values using a round-robin approach. For numerical attributes, this uses Bayesian ridge regression (Tipping, 2001), initialised with mean imputation, while for categorical attributes, we use logistic regression, initialised with mode imputation.

The scikit-learn implementations of nearest neighbour and iterative imputation can currently only impute numerical features, so we had to adapt them for categorical imputation. In all other aspects, we follow the default settings of scikit-learn.[6]

### 3.4 Classification Algorithms

We consider the classification algorithms listed in Table 1, as implemented in scikit-learn. Hyperparameters take their default values, except for SVM-L, LR and MLP, where we increase the maximum number of iterations to 10 000 to increase the probability of convergence.

For a number of these algorithms, specific ways have been proposed to handle missing values: e.g. NN-2-D (Dixon, 1979), SVM-G (Śmieja et al., 2019), MLP (Tresp et al., 1994; Śmieja et al., 2018; Ipsen et al., 2020)

---

[6]For the *nomao* dataset, iterative imputation diverged, so we had to restrict imputation to the interval $[-100, 100]$.

Table 2: Real-life classification datasets with missing values from the UCI repository for machine learning.

| Dataset | Records | Classes | Attributes | | | Missing value rate | | | Source |
|---|---|---|---|---|---|---|---|---|---|
| | | | Num | Cat | Total | Num | Cat | Total | |
| adult | 48 842 | 2 | 5 | 8 | 13 | 0.0 | 0.017 | 0.010 | Kohavi (1996) |
| agaricus-lepiota | 8124 | 2 | 2 | 20 | 22 | 0.0 | 0.015 | 0.014 | Schlimmer (1987) |
| aps-failure | 76 000 | 2 | 170 | 0 | 170 | 0.083 | | 0.083 | Ferreira Costa & Nascimento (2016) |
| arrhythmia | 452 | 13 | 279 | 0 | 279 | 0.0032 | | 0.0032 | Güvenir et al. (1997) |
| bands | 540 | 2 | 19 | 15 | 34 | 0.054 | 0.054 | 0.054 | Evans & Fisher (1994) |
| ckd | 400 | 2 | 24 | 0 | 24 | 0.11 | | 0.11 | Rubini & Eswaran (2015) |
| crx | 690 | 2 | 6 | 9 | 15 | 0.0060 | 0.0068 | 0.0065 | Quinlan (1987) |
| dress-sales | 500 | 2 | 3 | 9 | 12 | 0.20 | 0.19 | 0.19 | |
| exasens | 399 | 4 | 7 | 0 | 7 | 0.43 | | 0.43 | Soltani Zarrin et al. (2020) |
| hcc | 165 | 2 | 49 | 0 | 49 | 0.10 | | 0.10 | Santos et al. (2015) |
| heart-disease | 1611 | 2 | 13 | 1 | 14 | 0.18 | 0.0 | 0.17 | Detrano et al. (1989) |
| hepatitis | 155 | 2 | 19 | 0 | 19 | 0.057 | | 0.057 | Efron & Gong (1981) |
| horse-colic | 368 | 2 | 19 | 1 | 20 | 0.25 | 0.39 | 0.26 | McLeish & Cecile (1990) |
| mammographic-masses | 961 | 2 | 2 | 2 | 4 | 0.042 | 0.041 | 0.042 | Elter et al. (2007) |
| mi | 1700 | 8 | 111 | 0 | 111 | 0.085 | | 0.085 | Golovenkin et al. (2020) |
| nomao | 34 465 | 2 | 89 | 29 | 118 | 0.38 | 0.37 | 0.38 | Candillier & Lemaire (2012) |
| primary-tumor | 339 | 21 | 16 | 1 | 17 | 0.029 | 0.20 | 0.039 | Cestnik et al. (1987) |
| secom | 1567 | 2 | 590 | 0 | 590 | 0.045 | | 0.045 | McCann et al. (2008) |
| soybean | 683 | 19 | 22 | 13 | 35 | 0.099 | 0.096 | 0.098 | Michalski & Chilausky (1980) |
| thyroid0387 | 9172 | 18 | 7 | 22 | 29 | 0.22 | 0.0015 | 0.055 | Quinlan et al. (1986) |

and CART (Quinlan, 1989; Twala et al., 2008). The purpose of the present experiment is to evaluate the general approach of using imputation with missing-indicators when these solutions have not been implemented, as is the case in scikit-learn.

## 3.5 Datasets

We use twenty real-life datasets with naturally occurring missing values from the UCI repository for machine learning (Dua & Graff, 2019) (Table 2). We have preprocessed these datasets in the following manner. When it was clear from the description that an attribute was categorical, we treated it as such, even if it was originally represented with numerals. Conversely, where the possible values of an attribute admitted a semantic order, we encoded them numerically. We have removed attributes that were labelled non-informative by the accompanying documentation, as well as identifiers and alternative target values. For *dress-sales* (which does not appear to have been used in any publication), we cleaned up the data by eliminating spelling variations. For *heart-disease*, we reduced the id-attribute to only identify the source hospital. In *thyroid0387*, a small number of records belonged to multiple classes, which we resolved by retaining the most specific class.

## 4 Results

Using the experimental setup detailed in the previous section, we now try to answer the questions listed in Subsection 3.1. Note that fuller results are listed in the Appendix.

### 4.1 Do missing-indicators increase performance, and does it matter which imputation strategy they are paired with?

The *p*-values obtained by comparing imputation with and without missing-indicators are displayed in Table 3. Missing-indicators generally lead to increased performance — with the notable exception of CART, to which we return below.

Next, we consider how useful the more complicated imputation strategies are with respect to mean/mode imputation when we pair imputation with missing-indicators. Not very much, it turns out (Table 4). At best, nearest neighbour and iterative imputation only lead to a modest improvement, and for many classifiers,

Table 3: One-sided $p$-values, imputation with missing-indicators vs without.

| Classifier | Imputation strategy | | |
| --- | --- | --- | --- |
| | Mean/mode | Neighbours | Iterative |
| NN-1 | 0.024 | 0.0027 | 0.0011 |
| NN-2 | 0.035 | 0.0050 | 0.00085 |
| NN-1-D | 0.016 | 0.0031 | 0.00085 |
| NN-2-D | 0.0063 | 0.0070 | 0.00042 |
| SVM-L | 0.18 | 0.31 | 0.11 |
| SVM-G | 0.0063 | 0.0063 | 0.0027 |
| LR | 0.092 | 0.079 | 0.074 |
| MLP | 0.0050 | 0.013 | 0.011 |
| CART | 0.84 | 0.75 | 0.70 |
| RF | 0.058 | 0.12 | 0.29 |
| ERT | 0.36 | 0.018 | 0.027 |
| ABT | 0.089 | 0.10 | 0.49 |
| GBM | 0.39 | 0.022 | 0.18 |

Table 4: One-sided $p$-values, missing-indicators with iterative and nearest neighbour vs mean/mode imputation.

| Classifier | Imputation strategy | |
| --- | --- | --- |
| | Neighbours | Iterative |
| NN-1 | 0.90 | 0.27 |
| NN-2 | 0.74 | 0.26 |
| NN-1-D | 0.95 | 0.71 |
| NN-2-D | 0.80 | 0.34 |
| SVM-L | 0.48 | 0.61 |
| SVM-G | 0.47 | 0.94 |
| LR | 0.36 | 0.85 |
| MLP | 0.29 | 0.56 |
| CART | 0.67 | 0.69 |
| RF | 0.63 | 0.86 |
| ERT | 0.47 | 0.51 |
| ABT | 0.63 | 0.94 |
| GBM | 0.94 | 0.83 |

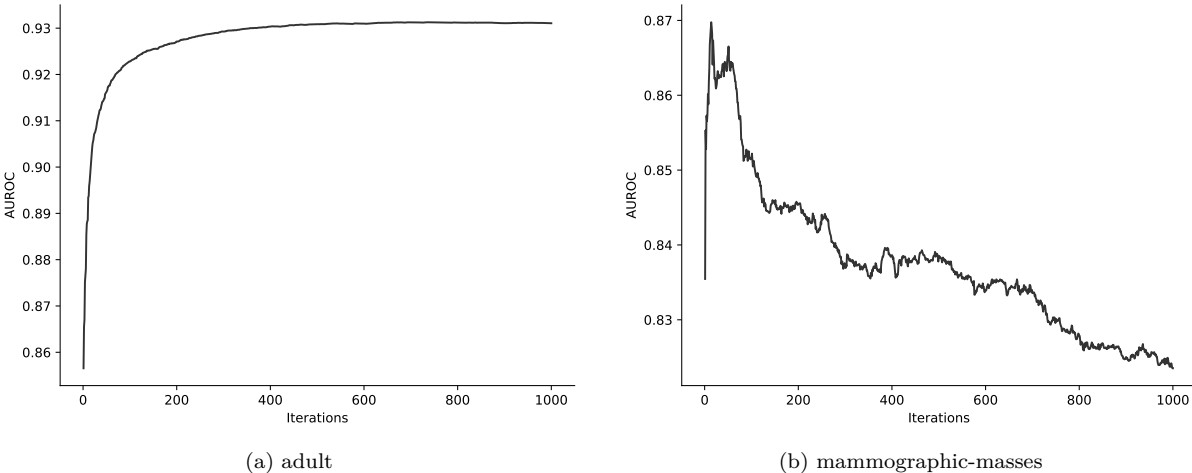

Figure 1: Test AUROC for GBM for two illustrative datasets, using mean/mode imputation without missing-indicators, for one random state and one cross-validation fold.

they actually decrease performance. Therefore, we focus on mean/mode imputation for the remainder of this section.

A possible reason for the poor performance of missing-indicators with CART, is that by default, the scikit-learn implementation of this classifier does not perform pruning, making it prone to overfitting. To test this hypothesis, we repeat our experiment for CART and mean imputation, but this time we apply cost complexity pruning ($\alpha = 0.01$). The resulting AUROC scores are now much better than the original scores without missing-indicators ($p = 0.013$) and somewhat better than cost complexity pruning without missing-indicators ($p = 0.23$).

In the case of ERT, missing-indicators may not lead to a clear performance improvement because of under-fitting. If we increase the number of trees from the default 100 to 1000, the improvement becomes somewhat clearer ($p = 0.15$).

For GBM, the default choice of 100 iterations of gradient descent can lead to both under- or overfitting, depending on the dataset (Fig. 1). We believe that it is generally preferable to continue training until an early-stopping criterion is met. If we apply the same criterion as with MLP,[7] the performance increase due to missing-indicators also becomes clearer ($p = 0.19$).

### 4.2 When do missing-indicators start to become useful in terms of missingness?

The theoretical motivation for representing missing values through missing-indicators is that this allows classifiers to learn the information encoded in their distribution. In general, this should be easier when there are more examples to learn from. If true, we can use this to obtain a better understanding of when missing-indicators might be useful on a per-attribute level.

We test this with the following additional experiment. For each attribute with missing values in each dataset, we reduce the original dataset by removing all other attributes with missing values. We thus obtain 1148 derived datasets, on which we again apply each of our classifiers (with pruning for CART, 1000 trees for ERT and early-stopping for GBM) and consider whether missing-indicators increase or decrease AUROC (we dismiss ties). Finally, for each classifier we fit a logistic regression model with cluster robust covariance (clustered by the originating dataset), with the following potential parameters: categoricalness (whether the

---

[7]Setting aside 10% of the data for validation, stopping when validation loss has not decreased by at least 0.0001 for ten iterations, with a maximum of 10 000 iterations.

Table 5: Thresholds above which missing-indicators are more likely than not to increase AUROC, in terms of the absolute number of missing values or the missing rate

| Classifier | Missing values | | Missing rate | |
| | Categorical | Numerical | Categorical | Numerical |
| --- | --- | --- | --- | --- |
| NN-1 | 20 | 296 | | |
| NN-2 | 10 | 146 | | |
| NN-1-D | 21 | 370 | | |
| NN-2-D | 6 | 80 | | |
| SVM-L | | | 0.0 | 0.0 |
| SVM-G | | | 0.0 | 0.44 |
| LR | | | 0.0 | 0.0 |
| CART | 1 | 13 | | |
| ERT | | | 0.0 | 1.0 |
| ABT | 1 | 18850 | | |
| GBM | | | 0.0 | 0.0 |

attribute is categorical) and either the number of missing values (log-transformed) or the missing rate. We use the Akaike information criterion (Akaike, 1971) to decide whether to select these parameters.

We find that for most classifiers, either the absolute or the relative number of missing values is an informative parameter with positive coefficient. For MLP, neither parameter is informative, while for RF, the number of missing values is an informative parameter with negative coefficient, for which we have no explanation at present. For every classifier except NN-1, NN-1-D and LR, categoricalness is an informative parameter with positive coefficient, meaning that missing-indicators are more beneficial for categorical than for numerical attributes.

The fitted logistic regression models allow us to calculate attribute-specific thresholds above which missing-indicators are more likely than not to increase AUROC, for all classifiers except MLP and RF (Table 5). In many cases, these thresholds are 1 or 0.0, indicating that missing-indicators are always likely to increase AUROC. We have included NN-1 and NN-1-D in this table on the basis of a model that includes the categoricalness parameter, since we find it implausible that it should not be relevant only for these specific classifiers. If we exclude it, the thresholds respectively become 233 and 285 records for categorical and numerical attributes alike. For LR, the threshold is a missing rate of 0.0, whether we include the categoricalness parameter or not.

### 4.3 Does using mean imputation instead of mode imputation allow for more information to be learned from missing categorical values?

As indicated above, missing-indicators are generally more likely to increase performance for categorical than for numerical attributes. A potential explanation for this is the fact that the mode of a categorical attribute is one of the non-missing values, whereas the mean of a numerical attribute is generally not equal to one of the non-missing values. Therefore, categorical imputation renders missing values truly indistinguishable from non-missing values, whereas numerical imputation does not — the information expressed by missing values may be partially recoverable, as argued by Le Morvan et al. (2021) and discussed in the Introduction.

We can achieve a similar partial representation of missing categorical values by changing the order in which we perform imputation and one-hot encoding, i.e. by performing numerical imputation on one-hot encoded categorical attributes with missing values. For imputation without missing-indicators, this indeed leads to better performance for some classifiers, while in combination with missing-indicators, it does not make much of a difference (Table 6)[8].

---

[8]LR is an exception here. We have no explanation for this, although we note that it corresponds with our finding in Subsection 4.2 that categoricalness is not a relevant factor for LR.

Table 6: One-sided $p$-values, mean imputation after one-hot encoding vs mode imputation of missing categorical values.

| Classifier | Without missing-indicators | With missing-indicators |
|---|---|---|
| NN-1 | 0.030 | 0.22 |
| NN-2 | 0.32 | 0.17 |
| NN-1-D | 0.030 | 0.36 |
| NN-2-D | 0.29 | 0.17 |
| SVM-L | 0.44 | 0.71 |
| SVM-G | 0.22 | 0.56 |
| LR | 0.88 | 0.023 |
| MLP | 0.14 | 0.52 |
| CART | 0.50 | 0.34 |
| RF | 0.084 | 0.78 |
| ERT | 0.023 | 0.95 |
| ABT | 0.56 | 0.66 |
| GBM | 0.12 | 0.56 |

We finish this section with a note of caution: because we have performed multiple statistical tests, it is quite likely that some particular results do not generalise to other datasets, without necessarily changing the overall picture. It is difficult to say much more about this, because the tests are not independent of each other.

## 5 Conclusion

We have presented the first large-scale experimental evaluation of the effect of the missing-indicator approach on classification performance, conducted on real datasets with naturally occurring missing values, paired with three different imputation techniques. The central question was whether, on balance, more benefit can be derived from the additional information encoded in a representation of missing values, or from the lower-dimensional projection of the data obtained by omitting missing-indicators.

We found that, on the whole, missing-indicators increase performance for the classification algorithms that we considered, although we cannot be sure for each classifier that this result will generalise to other datasets. The only classifier for which missing-indicators decreased performance was CART. We argued that this is due to overfitting by the default configuration of the scikit-learn implementation of CART, and showed that missing-indicators do increase performance when pruning is applied. For ERT and GBM, we were able to show that the advantage of including missing-indicators becomes more significant when the number of trees of ERT is increased to limit underfitting, and the number of iterations of GBM is determined dynamically by an early-stopping criterion to avoid both under- and overfitting.

We also looked at the relative performance of the three different imputation strategies and found that, in the presence of missing-indicators, nearest neighbour and iterative imputation do not increase performance over simple mean/mode imputation, with the possible exception of NN-2 and NN-2-D in the case of iterative imputation. This is a useful finding, because implementations of more sophisticated imputation strategies may not always be available to practitioners working in different frameworks, or easy to apply.

In a follow-up experiment, we were able to determine attribute-specific missingness thresholds above which missing-indicators are more likely than not to increase performance. We found that for categorical attributes, this threshold is generally very low, while for numerical attributes, there is more variation among classifiers, in particular as to whether this threshold is absolute or relative to the total number of records.

A possible explanation for the finding that missing-indicators are more useful for categorical than for numerical attributes is the fact that the mean of a numerical attribute is not generally identical to any of the non-missing values, and that mean imputation therefore preserves some of the information of missing

values. This is supported by the results of a further experiment, which showed that in the absence of missing-indicators, applying mean imputation to one-hot encoded categorical attributes results in somewhat better performance than mode imputation.

On the basis of the experiments in this paper, we conclude that the combination of mean/mode imputation with missing-indicators is a safe default approach towards missing values in classification tasks. While over- or underfitting is a concern for certain classifiers, it is a concern for these classifiers with or without missing-indicators. However, practitioners may want to omit missing-indicators when the classification algorithm to be used has a special provision for missing values, when the missingness thresholds that we determined are not met, or on the basis of specific information about the distribution of missing values in the dataset. The use of missing-indicators can also be combined with feature selection and dimensionality reduction algorithms to increase the information density of the resulting dataset.

While we have considered the use of missing-indicators with imputation, they can in principle also be used to supplement other, learner-specific solutions for missing-values. Whether this makes sense and increases performance will differ from case to case, and we leave this as an open question. In any case, we believe that going forward, any experimental evaluation of such learner-specific proposals should take missing-indicators into account.

The problem of missing data has been the subject of a rich body of theoretical literature. We hope to have contributed with this paper to the practical evaluation of some of that theory. In particular, we are happy to have identified twenty real-life datasets with missing values, and hope that in the future, more such datasets will be collected, which would allow drawing even firmer conclusions.

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

## A  Appendix

We list here the results of our experiments in greater detail. Table 7 contains the mean AUROC across five-fold cross-validation and five random states for each classifier, each dataset, each imputation strategy, without and with missing-indicators. Table 8 contains the mean AUROC for CART, GBM and ERT with updated hyperparameter values (as discussed in Subsection 4.1). Table 9 contains the mean AUROC obtained by imputing missing categorical values with the mean, after one-hot encoding (Subsection 4.3).

Table 7: AUROC, main experiment. **Bold**: higher value (without or with missing-indicators).

| Classifier | Dataset | Imputation strategy, missing-indicators no/yes | | | | | |
| | | Mean/mode | | Neighbours | | Iterative | |
| | | No | Yes | No | Yes | No | Yes |
| --- | --- | --- | --- | --- | --- | --- | --- |
| NN-1 | adult | 0.857 | **0.858** | 0.858 | **0.858** | **0.858** | 0.858 |
| | agaricus-lepiota | **1.000** | **1.000** | **1.000** | **1.000** | **1.000** | **1.000** |
| | aps-failure | **0.928** | 0.926 | **0.926** | 0.922 | **0.928** | 0.923 |
| | arrhythmia | **0.760** | 0.760 | **0.760** | 0.760 | 0.760 | **0.760** |
| | bands | 0.836 | **0.838** | 0.834 | **0.847** | 0.836 | **0.848** |
| | ckd | **0.998** | 0.994 | 0.991 | **0.993** | 0.989 | **0.991** |
| | crx | 0.908 | **0.909** | 0.904 | **0.908** | 0.909 | **0.910** |
| | dress-sales | 0.548 | **0.555** | 0.540 | **0.545** | 0.527 | **0.531** |
| | exasens | 0.710 | **0.726** | 0.703 | **0.713** | 0.717 | **0.726** |
| | hcc | 0.699 | **0.760** | 0.707 | **0.745** | 0.712 | **0.753** |
| | heart-disease | 0.846 | **0.847** | 0.841 | **0.844** | 0.843 | **0.846** |
| | hepatitis | **0.849** | 0.841 | 0.841 | **0.850** | 0.839 | **0.847** |
| | horse-colic | 0.716 | **0.733** | **0.738** | 0.734 | 0.726 | **0.738** |
| | mammographic-masses | 0.821 | **0.827** | 0.821 | **0.825** | 0.824 | **0.831** |
| | mi | 0.572 | **0.579** | 0.564 | **0.580** | 0.569 | **0.579** |
| | nomao | **0.983** | 0.982 | 0.978 | **0.981** | **0.983** | 0.982 |
| | primary-tumor | 0.675 | **0.687** | 0.678 | **0.693** | 0.676 | **0.687** |
| | secom | 0.641 | **0.651** | 0.641 | **0.643** | 0.646 | **0.653** |
| | soybean | 0.993 | **0.993** | 0.992 | **0.993** | 0.993 | **0.993** |
| | thyroid0387 | **0.879** | 0.877 | **0.878** | 0.877 | 0.875 | **0.875** |
| NN-2 | adult | 0.860 | **0.861** | 0.861 | **0.861** | **0.861** | 0.860 |
| | agaricus-lepiota | **1.000** | **1.000** | **1.000** | **1.000** | **1.000** | **1.000** |
| | aps-failure | 0.920 | **0.922** | 0.918 | **0.920** | 0.921 | **0.921** |
| | arrhythmia | 0.733 | **0.733** | **0.734** | 0.734 | 0.733 | **0.733** |
| | bands | 0.830 | **0.832** | 0.818 | **0.835** | 0.825 | **0.836** |
| | ckd | **0.999** | 0.996 | 0.992 | **0.995** | 0.991 | **0.993** |
| | crx | 0.899 | **0.900** | 0.898 | **0.899** | 0.900 | **0.901** |
| | dress-sales | **0.554** | 0.547 | **0.541** | 0.539 | **0.532** | 0.527 |
| | exasens | 0.709 | **0.716** | 0.699 | **0.706** | 0.712 | **0.718** |
| | hcc | 0.690 | **0.696** | 0.695 | **0.709** | 0.698 | **0.705** |

Table 7: AUROC, main experiment. **Bold**: higher value (without or with missing-indicators).

| Classifier | Dataset | Imputation strategy, missing-indicators no/yes | | | | | |
| | | Mean/mode | | Neighbours | | Iterative | |
| | | No | Yes | No | Yes | No | Yes |
|---|---|---|---|---|---|---|---|
| | heart-disease | 0.831 | **0.835** | 0.828 | **0.837** | 0.829 | **0.836** |
| | hepatitis | **0.861** | 0.851 | 0.846 | **0.850** | 0.860 | **0.862** |
| | horse-colic | 0.684 | **0.710** | **0.724** | 0.706 | 0.695 | **0.704** |
| | mammographic-masses | 0.820 | **0.825** | 0.821 | **0.824** | 0.822 | **0.828** |
| | mi | 0.561 | **0.563** | 0.555 | **0.560** | 0.563 | **0.563** |
| | nomao | 0.980 | **0.982** | 0.976 | **0.980** | 0.980 | **0.981** |
| | primary-tumor | 0.667 | **0.673** | 0.670 | **0.675** | 0.666 | **0.677** |
| | secom | 0.607 | **0.612** | 0.614 | **0.617** | 0.607 | **0.613** |
| | soybean | 0.986 | **0.988** | 0.987 | **0.988** | 0.986 | **0.988** |
| | thyroid0387 | **0.878** | 0.877 | **0.878** | 0.876 | **0.871** | 0.871 |
| NN-1-D | adult | 0.838 | **0.838** | 0.837 | **0.839** | 0.837 | **0.838** |
| | agaricus-lepiota | **1.000** | **1.000** | **1.000** | **1.000** | **1.000** | **1.000** |
| | aps-failure | **0.929** | 0.926 | **0.927** | 0.922 | **0.928** | 0.923 |
| | arrhythmia | **0.764** | 0.764 | **0.763** | 0.763 | 0.764 | **0.764** |
| | bands | 0.871 | **0.875** | 0.865 | **0.879** | 0.870 | **0.880** |
| | ckd | **0.998** | 0.994 | 0.991 | **0.993** | 0.989 | **0.991** |
| | crx | 0.907 | **0.908** | 0.905 | **0.908** | 0.908 | **0.909** |
| | dress-sales | 0.544 | **0.560** | 0.538 | **0.545** | 0.528 | **0.535** |
| | exasens | 0.629 | **0.641** | 0.625 | **0.634** | 0.632 | **0.640** |
| | hcc | 0.728 | **0.786** | 0.733 | **0.772** | 0.738 | **0.773** |
| | heart-disease | 0.847 | **0.848** | 0.843 | **0.845** | 0.843 | **0.847** |
| | hepatitis | **0.857** | 0.853 | 0.841 | **0.855** | 0.841 | **0.853** |
| | horse-colic | 0.743 | **0.751** | **0.762** | 0.752 | 0.749 | **0.757** |
| | mammographic-masses | 0.802 | **0.806** | 0.798 | **0.805** | 0.803 | **0.808** |
| | mi | 0.572 | **0.580** | 0.564 | **0.580** | 0.569 | **0.579** |
| | nomao | **0.984** | 0.983 | 0.979 | **0.982** | **0.984** | 0.983 |
| | primary-tumor | 0.665 | **0.676** | 0.667 | **0.684** | 0.665 | **0.677** |
| | secom | 0.644 | **0.652** | 0.644 | **0.645** | 0.647 | **0.655** |
| | soybean | 0.993 | **0.993** | 0.992 | **0.993** | 0.993 | 0.993 |
| | thyroid0387 | **0.881** | 0.879 | **0.880** | 0.879 | **0.877** | 0.877 |
| NN-2-D | adult | 0.842 | **0.843** | 0.842 | **0.843** | 0.842 | **0.843** |
| | agaricus-lepiota | **1.000** | **1.000** | **1.000** | **1.000** | **1.000** | **1.000** |
| | aps-failure | 0.920 | **0.922** | 0.918 | **0.921** | 0.922 | **0.922** |
| | arrhythmia | 0.735 | **0.736** | **0.736** | 0.736 | 0.735 | **0.735** |
| | bands | 0.859 | **0.861** | 0.844 | **0.863** | 0.850 | **0.863** |
| | ckd | **0.999** | 0.996 | 0.992 | **0.995** | 0.991 | **0.993** |
| | crx | 0.898 | **0.899** | 0.898 | **0.900** | 0.900 | **0.901** |
| | dress-sales | 0.548 | **0.548** | 0.543 | 0.538 | **0.534** | 0.532 |
| | exasens | 0.628 | **0.635** | 0.623 | **0.629** | 0.629 | **0.634** |
| | hcc | 0.710 | **0.723** | 0.716 | **0.737** | 0.719 | **0.729** |
| | heart-disease | 0.833 | **0.838** | 0.830 | **0.839** | 0.831 | **0.839** |
| | hepatitis | **0.862** | 0.856 | 0.847 | **0.852** | 0.859 | **0.865** |
| | horse-colic | 0.712 | **0.731** | **0.745** | 0.730 | 0.719 | **0.729** |
| | mammographic-masses | 0.802 | **0.805** | 0.799 | **0.804** | 0.802 | **0.807** |
| | mi | 0.560 | **0.563** | 0.556 | **0.560** | 0.564 | **0.565** |
| | nomao | 0.981 | **0.983** | 0.977 | **0.981** | 0.981 | **0.982** |
| | primary-tumor | 0.659 | **0.666** | 0.660 | **0.667** | 0.657 | **0.669** |

Table 7: AUROC, main experiment. **Bold**: higher value (without or with missing-indicators).

| Classifier | Dataset | Imputation strategy, missing-indicators no/yes | | | | | |
| | | Mean/mode | | Neighbours | | Iterative | |
| | | No | Yes | No | Yes | No | Yes |
|---|---|---|---|---|---|---|---|
| | secom | 0.606 | **0.610** | 0.612 | **0.615** | 0.606 | **0.611** |
| | soybean | 0.986 | **0.988** | 0.987 | **0.988** | 0.986 | **0.988** |
| | thyroid0387 | **0.880** | 0.878 | **0.879** | 0.878 | **0.872** | 0.872 |
| SVM-L | adult | 0.905 | **0.906** | 0.905 | **0.906** | 0.905 | **0.906** |
| | agaricus-lepiota | **1.000** | **1.000** | **1.000** | **1.000** | **1.000** | **1.000** |
| | aps-failure | 0.966 | **0.969** | 0.961 | **0.969** | 0.963 | **0.966** |
| | arrhythmia | 0.818 | **0.843** | 0.819 | **0.843** | 0.818 | **0.843** |
| | bands | 0.796 | **0.817** | 0.791 | **0.809** | 0.760 | **0.801** |
| | ckd | 1.000 | **1.000** | 0.999 | **1.000** | 0.999 | **1.000** |
| | crx | **0.922** | 0.920 | **0.920** | 0.920 | **0.922** | 0.921 |
| | dress-sales | **0.598** | 0.593 | **0.594** | 0.588 | 0.591 | **0.597** |
| | exasens | 0.762 | **0.780** | 0.761 | **0.769** | 0.761 | **0.780** |
| | hcc | **0.757** | 0.738 | **0.781** | 0.756 | **0.746** | 0.733 |
| | heart-disease | **0.866** | 0.865 | 0.866 | **0.867** | 0.867 | **0.868** |
| | hepatitis | **0.848** | 0.824 | **0.857** | 0.831 | **0.856** | 0.833 |
| | horse-colic | **0.790** | 0.784 | **0.798** | 0.784 | **0.770** | 0.762 |
| | mammographic-masses | 0.865 | **0.867** | 0.862 | **0.865** | **0.864** | 0.864 |
| | mi | 0.641 | **0.666** | 0.639 | **0.669** | 0.636 | **0.671** |
| | nomao | 0.986 | **0.988** | 0.986 | **0.988** | 0.985 | **0.988** |
| | primary-tumor | **0.769** | 0.769 | **0.772** | 0.770 | **0.778** | 0.777 |
| | secom | 0.626 | **0.629** | **0.671** | 0.659 | **0.631** | 0.628 |
| | soybean | 0.999 | **0.999** | 0.999 | **0.999** | 0.999 | **0.999** |
| | thyroid0387 | 0.964 | **0.964** | 0.964 | **0.964** | 0.954 | **0.956** |
| SVM-G | adult | 0.895 | **0.897** | 0.896 | **0.896** | 0.896 | **0.897** |
| | agaricus-lepiota | **1.000** | **1.000** | **1.000** | **1.000** | **1.000** | **1.000** |
| | aps-failure | 0.967 | **0.968** | 0.960 | **0.965** | 0.965 | **0.966** |
| | arrhythmia | 0.848 | **0.848** | **0.848** | 0.848 | 0.848 | **0.848** |
| | bands | 0.855 | **0.865** | 0.858 | **0.870** | 0.857 | **0.869** |
| | ckd | 1.000 | **1.000** | 1.000 | **1.000** | 1.000 | **1.000** |
| | crx | 0.926 | **0.927** | 0.924 | **0.927** | 0.926 | **0.928** |
| | dress-sales | 0.618 | **0.620** | **0.620** | 0.619 | 0.607 | **0.612** |
| | exasens | 0.772 | **0.780** | 0.767 | **0.780** | 0.773 | **0.780** |
| | hcc | 0.778 | **0.790** | 0.785 | **0.793** | 0.770 | **0.783** |
| | heart-disease | **0.865** | 0.864 | 0.863 | **0.864** | 0.864 | **0.864** |
| | hepatitis | **0.893** | 0.892 | **0.888** | 0.887 | **0.893** | 0.890 |
| | horse-colic | 0.768 | **0.771** | 0.784 | **0.786** | 0.767 | **0.769** |
| | mammographic-masses | 0.840 | **0.845** | 0.838 | **0.841** | 0.839 | **0.842** |
| | mi | 0.635 | **0.643** | 0.637 | **0.645** | 0.639 | **0.648** |
| | nomao | 0.991 | **0.992** | 0.988 | **0.991** | 0.989 | **0.991** |
| | primary-tumor | 0.762 | **0.765** | 0.764 | **0.767** | 0.766 | **0.767** |
| | secom | **0.699** | 0.694 | **0.702** | 0.698 | **0.689** | 0.685 |
| | soybean | 0.999 | **0.999** | 0.999 | **0.999** | 0.999 | **0.999** |
| | thyroid0387 | **0.978** | 0.978 | **0.978** | 0.977 | 0.969 | **0.970** |
| LR | adult | 0.905 | **0.906** | 0.906 | **0.906** | 0.906 | **0.906** |
| | agaricus-lepiota | **1.000** | **1.000** | **1.000** | **1.000** | **1.000** | **1.000** |
| | aps-failure | 0.971 | **0.979** | 0.971 | **0.980** | 0.967 | **0.978** |
| | arrhythmia | 0.860 | **0.860** | 0.860 | **0.860** | 0.859 | **0.860** |

Table 7: AUROC, main experiment. **Bold**: higher value (without or with missing-indicators).

| Classifier | Dataset | Imputation strategy, missing-indicators no/yes | | | | | |
| | | Mean/mode | | Neighbours | | Iterative | |
| | | No | Yes | No | Yes | No | Yes |
|---|---|---|---|---|---|---|---|
| | bands | 0.819 | **0.833** | 0.811 | **0.830** | 0.808 | **0.828** |
| | ckd | 1.000 | **1.000** | 1.000 | **1.000** | 1.000 | **1.000** |
| | crx | **0.924** | 0.923 | 0.923 | **0.923** | **0.924** | 0.924 |
| | dress-sales | 0.620 | **0.620** | 0.619 | **0.624** | 0.614 | **0.620** |
| | exasens | 0.774 | **0.783** | 0.768 | **0.775** | 0.773 | **0.782** |
| | hcc | **0.778** | 0.760 | **0.796** | 0.774 | **0.772** | 0.755 |
| | heart-disease | 0.867 | **0.868** | 0.867 | **0.869** | 0.867 | **0.869** |
| | hepatitis | **0.863** | 0.856 | **0.871** | 0.862 | **0.870** | 0.862 |
| | horse-colic | **0.789** | 0.786 | **0.793** | 0.786 | **0.769** | 0.764 |
| | mammographic-masses | 0.866 | **0.868** | 0.863 | **0.865** | **0.865** | 0.865 |
| | mi | 0.654 | **0.685** | 0.645 | **0.685** | 0.650 | **0.688** |
| | nomao | 0.986 | **0.988** | 0.986 | **0.988** | 0.985 | **0.988** |
| | primary-tumor | 0.773 | **0.776** | 0.772 | **0.775** | 0.780 | **0.783** |
| | secom | **0.686** | 0.678 | **0.687** | 0.680 | **0.676** | 0.673 |
| | soybean | 0.999 | **0.999** | 0.999 | **0.999** | 0.999 | **0.999** |
| | thyroid0387 | 0.974 | **0.975** | 0.975 | **0.975** | 0.973 | **0.973** |
| MLP | adult | 0.890 | **0.890** | 0.891 | 0.889 | **0.891** | 0.890 |
| | agaricus-lepiota | **1.000** | 1.000 | **1.000** | 1.000 | **1.000** | 1.000 |
| | aps-failure | 0.928 | **0.942** | 0.931 | **0.943** | 0.931 | **0.942** |
| | arrhythmia | 0.831 | **0.846** | 0.831 | **0.845** | 0.831 | **0.845** |
| | bands | 0.871 | **0.879** | 0.873 | **0.885** | 0.868 | **0.882** |
| | ckd | 1.000 | **1.000** | 1.000 | **1.000** | 0.999 | **1.000** |
| | crx | 0.902 | **0.906** | 0.901 | **0.905** | 0.900 | **0.905** |
| | dress-sales | 0.549 | **0.553** | 0.560 | **0.561** | 0.544 | **0.545** |
| | exasens | 0.759 | **0.762** | 0.746 | **0.755** | 0.757 | **0.763** |
| | hcc | 0.778 | **0.781** | 0.791 | **0.796** | 0.777 | **0.781** |
| | heart-disease | **0.819** | 0.815 | **0.816** | 0.811 | **0.818** | 0.816 |
| | hepatitis | **0.861** | 0.861 | **0.870** | 0.865 | **0.872** | 0.866 |
| | horse-colic | 0.714 | **0.744** | 0.727 | **0.756** | 0.719 | **0.734** |
| | mammographic-masses | **0.845** | 0.840 | **0.841** | 0.836 | **0.847** | 0.840 |
| | mi | 0.659 | **0.695** | 0.656 | **0.697** | 0.660 | **0.697** |
| | nomao | 0.991 | **0.991** | 0.987 | **0.990** | 0.990 | **0.991** |
| | primary-tumor | 0.768 | **0.782** | 0.765 | **0.778** | 0.769 | **0.785** |
| | secom | 0.693 | **0.701** | 0.699 | **0.704** | 0.686 | **0.697** |
| | soybean | 0.999 | **0.999** | 0.999 | 0.999 | 0.999 | **0.999** |
| | thyroid0387 | **0.988** | 0.988 | **0.988** | 0.987 | **0.986** | 0.985 |
| CART | adult | **0.776** | 0.775 | **0.776** | 0.775 | **0.776** | 0.774 |
| | agaricus-lepiota | **1.000** | 1.000 | **1.000** | 1.000 | **1.000** | 1.000 |
| | aps-failure | 0.855 | **0.858** | **0.858** | 0.857 | 0.854 | **0.857** |
| | arrhythmia | **0.712** | 0.710 | **0.712** | 0.702 | **0.714** | 0.702 |
| | bands | **0.716** | 0.713 | 0.697 | **0.716** | 0.706 | **0.717** |
| | ckd | **0.965** | 0.964 | **0.979** | 0.978 | **0.972** | 0.970 |
| | crx | **0.818** | 0.812 | **0.813** | 0.810 | **0.815** | 0.809 |
| | dress-sales | 0.524 | **0.548** | 0.526 | **0.529** | **0.534** | 0.532 |
| | exasens | **0.618** | 0.616 | **0.618** | 0.608 | 0.621 | **0.626** |
| | hcc | 0.593 | **0.603** | **0.619** | 0.617 | **0.614** | 0.601 |
| | heart-disease | 0.702 | **0.703** | **0.701** | 0.700 | 0.703 | **0.706** |

Continued on next page

Table 7: AUROC, main experiment. **Bold**: higher value (without or with missing-indicators).

| Classifier | Dataset | Mean/mode | | Neighbours | | Iterative | |
|---|---|---|---|---|---|---|---|
| | | No | Yes | No | Yes | No | Yes |
| | hepatitis | **0.660** | 0.657 | **0.691** | 0.673 | **0.703** | 0.700 |
| | horse-colic | **0.695** | 0.673 | **0.700** | 0.663 | **0.680** | 0.676 |
| | mammographic-masses | **0.748** | 0.744 | **0.747** | 0.746 | 0.744 | **0.746** |
| | mi | **0.572** | 0.572 | 0.549 | **0.574** | 0.557 | **0.571** |
| | nomao | **0.935** | 0.935 | 0.922 | **0.925** | 0.926 | **0.927** |
| | primary-tumor | 0.621 | **0.621** | 0.625 | **0.627** | 0.622 | **0.623** |
| | secom | 0.547 | **0.552** | 0.555 | **0.558** | **0.542** | 0.538 |
| | soybean | 0.975 | **0.977** | 0.973 | **0.974** | 0.971 | **0.973** |
| | thyroid0387 | **0.897** | 0.888 | **0.875** | 0.871 | **0.886** | 0.883 |
| RF | adult | **0.890** | 0.890 | 0.890 | **0.891** | **0.891** | 0.890 |
| | agaricus-lepiota | **1.000** | **1.000** | **1.000** | **1.000** | **1.000** | **1.000** |
| | aps-failure | 0.988 | **0.989** | 0.988 | **0.989** | **0.988** | 0.988 |
| | arrhythmia | 0.883 | **0.884** | **0.885** | 0.885 | **0.886** | 0.883 |
| | bands | 0.893 | **0.896** | 0.886 | **0.898** | 0.896 | **0.896** |
| | ckd | 1.000 | **1.000** | **1.000** | 1.000 | **1.000** | 1.000 |
| | crx | **0.932** | 0.931 | **0.934** | 0.932 | 0.931 | **0.931** |
| | dress-sales | 0.591 | **0.606** | 0.583 | **0.602** | 0.582 | **0.597** |
| | exasens | **0.701** | 0.701 | 0.689 | **0.694** | 0.698 | **0.701** |
| | hcc | 0.803 | **0.816** | **0.813** | 0.813 | 0.794 | **0.806** |
| | heart-disease | 0.861 | **0.864** | 0.862 | **0.866** | 0.864 | **0.866** |
| | hepatitis | 0.882 | **0.887** | **0.890** | 0.887 | **0.888** | 0.886 |
| | horse-colic | **0.800** | 0.791 | **0.811** | 0.809 | **0.793** | 0.792 |
| | mammographic-masses | 0.812 | **0.821** | 0.815 | **0.819** | 0.812 | **0.820** |
| | mi | **0.687** | 0.687 | 0.676 | **0.681** | **0.687** | 0.679 |
| | nomao | 0.994 | **0.994** | 0.991 | **0.992** | 0.993 | **0.993** |
| | primary-tumor | 0.749 | **0.758** | 0.730 | **0.761** | 0.748 | **0.761** |
| | secom | **0.722** | 0.710 | **0.719** | 0.713 | **0.722** | 0.710 |
| | soybean | **0.999** | 0.999 | 0.999 | **0.999** | **0.999** | 0.999 |
| | thyroid0387 | **0.994** | 0.994 | **0.993** | 0.992 | **0.995** | 0.992 |
| ERT | adult | 0.846 | **0.847** | 0.847 | **0.847** | 0.846 | **0.847** |
| | agaricus-lepiota | **1.000** | **1.000** | **1.000** | **1.000** | **1.000** | **1.000** |
| | aps-failure | 0.989 | **0.989** | **0.989** | 0.988 | 0.989 | **0.989** |
| | arrhythmia | 0.885 | **0.889** | 0.881 | **0.885** | 0.881 | **0.885** |
| | bands | 0.889 | **0.890** | 0.874 | **0.890** | 0.885 | **0.892** |
| | ckd | 1.000 | **1.000** | **1.000** | 1.000 | 1.000 | **1.000** |
| | crx | **0.913** | 0.911 | **0.916** | 0.915 | **0.912** | 0.910 |
| | dress-sales | 0.572 | **0.600** | 0.563 | **0.594** | 0.560 | **0.589** |
| | exasens | **0.633** | 0.632 | 0.622 | **0.626** | 0.624 | **0.630** |
| | hcc | 0.783 | **0.799** | 0.776 | **0.804** | 0.771 | **0.796** |
| | heart-disease | 0.858 | **0.861** | 0.862 | **0.865** | **0.861** | 0.861 |
| | hepatitis | **0.871** | 0.861 | 0.876 | **0.877** | **0.882** | 0.871 |
| | horse-colic | **0.793** | 0.780 | **0.818** | 0.796 | **0.790** | 0.780 |
| | mammographic-masses | 0.793 | **0.801** | 0.791 | **0.800** | 0.793 | **0.801** |
| | mi | **0.689** | 0.683 | 0.661 | **0.683** | 0.676 | **0.686** |
| | nomao | **0.994** | 0.993 | 0.991 | **0.992** | 0.993 | **0.993** |
| | primary-tumor | 0.702 | **0.718** | 0.698 | **0.717** | 0.704 | **0.721** |
| | secom | **0.718** | 0.713 | **0.716** | 0.705 | 0.706 | **0.716** |

Table 7: AUROC, main experiment. **Bold**: higher value (without or with missing-indicators).

| Classifier | Dataset | Imputation strategy, missing-indicators no/yes | | | | | |
| | | Mean/mode | | Neighbours | | Iterative | |
| | | No | Yes | No | Yes | No | Yes |
| --- | --- | --- | --- | --- | --- | --- | --- |
| ABT | soybean | **0.999** | 0.999 | 0.999 | **0.999** | **0.999** | 0.999 |
| | thyroid0387 | **0.979** | 0.976 | **0.980** | 0.979 | 0.975 | **0.977** |
| | adult | 0.915 | **0.915** | 0.915 | **0.915** | 0.915 | **0.915** |
| | agaricus-lepiota | **1.000** | **1.000** | **1.000** | **1.000** | **1.000** | **1.000** |
| | aps-failure | 0.987 | **0.987** | 0.987 | **0.987** | 0.986 | **0.987** |
| | arrhythmia | **0.634** | 0.632 | **0.634** | 0.633 | **0.634** | 0.632 |
| | bands | 0.806 | **0.806** | 0.793 | **0.809** | 0.805 | **0.807** |
| | ckd | 1.000 | **1.000** | 0.998 | **0.999** | 0.998 | **1.000** |
| | crx | 0.905 | **0.906** | **0.907** | 0.906 | **0.909** | 0.905 |
| | dress-sales | **0.590** | 0.582 | **0.584** | 0.578 | 0.587 | **0.589** |
| | exasens | **0.720** | **0.720** | 0.705 | **0.717** | **0.713** | 0.711 |
| | hcc | 0.715 | **0.724** | **0.739** | 0.735 | **0.708** | 0.687 |
| | heart-disease | 0.860 | **0.860** | 0.857 | **0.861** | **0.861** | 0.858 |
| | hepatitis | 0.797 | **0.804** | 0.824 | **0.830** | 0.805 | **0.814** |
| | horse-colic | **0.753** | 0.752 | **0.749** | 0.742 | **0.735** | 0.729 |
| | mammographic-masses | 0.856 | **0.857** | 0.855 | **0.856** | 0.854 | **0.855** |
| | mi | 0.555 | **0.572** | 0.572 | **0.586** | **0.573** | 0.572 |
| | nomao | 0.987 | **0.987** | 0.985 | **0.986** | 0.986 | **0.986** |
| | primary-tumor | **0.661** | 0.660 | **0.670** | 0.668 | 0.668 | **0.671** |
| | secom | 0.670 | **0.670** | **0.661** | **0.661** | 0.663 | **0.663** |
| | soybean | 0.863 | **0.871** | 0.777 | **0.850** | 0.855 | **0.865** |
| | thyroid0387 | **0.685** | **0.685** | **0.666** | **0.666** | **0.674** | **0.674** |
| GBM | adult | 0.921 | **0.921** | 0.921 | **0.921** | 0.921 | **0.921** |
| | agaricus-lepiota | **1.000** | **1.000** | **1.000** | **1.000** | **1.000** | **1.000** |
| | aps-failure | **0.989** | 0.988 | 0.988 | **0.989** | **0.988** | 0.988 |
| | arrhythmia | 0.873 | **0.874** | **0.880** | 0.875 | **0.879** | 0.878 |
| | bands | 0.869 | **0.870** | 0.857 | **0.871** | 0.870 | **0.873** |
| | ckd | 1.000 | **1.000** | 0.998 | **0.998** | 0.998 | **0.999** |
| | crx | **0.932** | 0.932 | 0.930 | **0.931** | 0.929 | **0.931** |
| | dress-sales | **0.612** | 0.606 | 0.597 | **0.601** | **0.612** | 0.609 |
| | exasens | **0.725** | 0.725 | 0.720 | **0.724** | 0.723 | **0.725** |
| | hcc | 0.759 | **0.780** | 0.762 | **0.773** | **0.747** | 0.742 |
| | heart-disease | 0.872 | **0.872** | 0.869 | **0.870** | **0.873** | 0.872 |
| | hepatitis | **0.837** | 0.828 | 0.837 | **0.838** | 0.854 | **0.854** |
| | horse-colic | **0.793** | 0.789 | **0.794** | 0.789 | **0.798** | 0.789 |
| | mammographic-masses | 0.850 | **0.853** | 0.847 | **0.851** | 0.846 | **0.853** |
| | mi | **0.664** | 0.663 | 0.659 | **0.663** | 0.654 | **0.661** |
| | nomao | 0.991 | **0.991** | 0.989 | **0.990** | 0.991 | **0.991** |
| | primary-tumor | 0.760 | **0.763** | 0.762 | **0.762** | **0.754** | 0.752 |
| | secom | 0.708 | **0.710** | **0.717** | 0.716 | 0.708 | **0.711** |
| | soybean | 0.999 | **0.999** | 0.998 | **0.999** | 0.998 | **0.998** |
| | thyroid0387 | **0.916** | 0.914 | **0.896** | 0.896 | 0.903 | **0.905** |

Table 8: AUROC, additional experiment for mean/mode imputation and classifiers with adjusted hyperparameter values. **Bold**: higher value (without or with missing-indicators).

| Dataset | Classifier, missing-indicators no/yes | | | | | |
| | CART | | GBM | | ERT | |
| | No | Yes | No | Yes | No | Yes |
|---|---|---|---|---|---|---|
| adult | **0.844** | **0.844** | 0.927 | **0.927** | 0.847 | **0.847** |
| agaricus-lepiota | 0.991 | **0.992** | **1.000** | **1.000** | **1.000** | **1.000** |
| aps-failure | **0.859** | **0.859** | 0.988 | **0.988** | **0.991** | 0.991 |
| arrhythmia | **0.749** | 0.748 | 0.850 | **0.852** | 0.897 | **0.899** |
| bands | 0.749 | **0.759** | 0.855 | **0.857** | 0.890 | **0.890** |
| ckd | **0.968** | 0.967 | **0.998** | 0.998 | 1.000 | **1.000** |
| crx | **0.897** | 0.897 | **0.934** | 0.933 | 0.914 | **0.914** |
| dress-sales | 0.568 | **0.570** | 0.608 | **0.614** | 0.572 | **0.602** |
| exasens | 0.723 | **0.732** | 0.755 | **0.757** | **0.626** | 0.626 |
| hcc | 0.577 | **0.588** | 0.737 | **0.745** | 0.791 | **0.808** |
| heart-disease | **0.777** | **0.777** | 0.870 | **0.871** | 0.861 | **0.862** |
| hepatitis | **0.626** | 0.578 | **0.812** | 0.809 | **0.877** | 0.873 |
| horse-colic | **0.742** | 0.724 | **0.789** | 0.783 | **0.799** | 0.782 |
| mammographic-masses | **0.823** | **0.823** | 0.857 | **0.859** | 0.795 | **0.802** |
| mi | 0.586 | **0.592** | **0.650** | 0.639 | **0.702** | 0.695 |
| nomao | **0.916** | **0.916** | **0.994** | 0.994 | **0.994** | 0.994 |
| primary-tumor | 0.703 | **0.707** | 0.766 | **0.767** | 0.705 | **0.714** |
| secom | **0.500** | **0.500** | **0.684** | 0.677 | 0.746 | **0.747** |
| soybean | 0.990 | **0.991** | **0.999** | 0.999 | **0.999** | 0.999 |
| thyroid0387 | **0.909** | **0.909** | 0.913 | **0.923** | **0.987** | 0.987 |

Table 9: AUROC, additional experiment for imputation of categorical attributes (mode imputation or mean imputation after one-hot encoding). **Bold**: higher value.

| Classifier | Dataset | Without missing-indicators | | With missing-indicators | |
| | | Mode | Mean | Mode | Mean |
|---|---|---|---|---|---|
| NN-1 | adult | 0.857 | **0.858** | 0.858 | **0.858** |
| | agaricus-lepiota | **1.000** | **1.000** | **1.000** | **1.000** |
| | bands | 0.836 | **0.839** | 0.838 | **0.843** |
| | crx | 0.908 | **0.909** | **0.909** | 0.909 |
| | dress-sales | **0.548** | 0.533 | **0.555** | 0.539 |
| | horse-colic | 0.716 | **0.737** | 0.733 | **0.737** |
| | mammographic-masses | 0.821 | **0.831** | 0.827 | **0.828** |
| | nomao | 0.983 | **0.984** | **0.982** | 0.982 |
| | primary-tumor | 0.675 | **0.679** | 0.687 | **0.693** |
| | soybean | 0.993 | **0.993** | 0.993 | **0.993** |
| | thyroid0387 | 0.879 | **0.879** | **0.877** | 0.877 |
| NN-2 | adult | 0.860 | **0.861** | 0.861 | **0.861** |
| | agaricus-lepiota | **1.000** | **1.000** | **1.000** | **1.000** |
| | bands | **0.830** | 0.829 | 0.832 | **0.834** |
| | crx | **0.899** | 0.898 | 0.900 | **0.900** |
| | dress-sales | **0.554** | 0.548 | **0.547** | 0.531 |
| | horse-colic | 0.684 | **0.688** | 0.710 | **0.719** |
| | mammographic-masses | 0.820 | **0.824** | 0.825 | **0.825** |

Table 9: AUROC, additional experiment for imputation of categorical attributes (mode imputation or mean imputation after one-hot encoding). **Bold**: higher value.

| Classifier | Dataset | Without missing-indicators | | With missing-indicators | |
|---|---|---|---|---|---|
| | | Mode | Mean | Mode | Mean |
| | nomao | 0.980 | **0.981** | 0.982 | **0.982** |
| | primary-tumor | 0.667 | **0.669** | 0.673 | **0.674** |
| | soybean | **0.986** | 0.986 | **0.988** | 0.988 |
| | thyroid0387 | 0.878 | **0.879** | **0.877** | 0.876 |
| NN-1-D | adult | 0.838 | **0.838** | **0.838** | 0.838 |
| | agaricus-lepiota | **1.000** | **1.000** | **1.000** | **1.000** |
| | bands | 0.871 | **0.874** | 0.875 | **0.876** |
| | crx | 0.907 | **0.908** | **0.908** | 0.908 |
| | dress-sales | **0.544** | 0.537 | **0.560** | 0.544 |
| | horse-colic | 0.743 | **0.763** | 0.751 | **0.756** |
| | mammographic-masses | 0.802 | **0.810** | 0.806 | **0.807** |
| | nomao | 0.984 | **0.985** | **0.983** | 0.983 |
| | primary-tumor | 0.665 | **0.669** | 0.676 | **0.681** |
| | soybean | 0.993 | **0.993** | 0.993 | **0.993** |
| | thyroid0387 | **0.881** | 0.880 | **0.879** | 0.879 |
| NN-2-D | adult | 0.842 | **0.843** | 0.843 | **0.843** |
| | agaricus-lepiota | **1.000** | **1.000** | **1.000** | **1.000** |
| | bands | **0.859** | 0.857 | 0.861 | **0.862** |
| | crx | **0.898** | 0.898 | 0.899 | **0.900** |
| | dress-sales | **0.548** | 0.543 | **0.548** | 0.535 |
| | horse-colic | 0.712 | **0.716** | 0.731 | **0.739** |
| | mammographic-masses | 0.802 | **0.806** | 0.805 | **0.806** |
| | nomao | 0.981 | **0.982** | 0.983 | **0.983** |
| | primary-tumor | 0.659 | **0.661** | 0.666 | **0.667** |
| | soybean | **0.986** | 0.986 | **0.988** | 0.988 |
| | thyroid0387 | 0.880 | **0.880** | **0.878** | 0.877 |
| SVM-L | adult | 0.905 | **0.905** | 0.906 | **0.906** |
| | agaricus-lepiota | **1.000** | **1.000** | **1.000** | **1.000** |
| | bands | 0.796 | **0.797** | 0.817 | **0.817** |
| | crx | **0.922** | 0.921 | **0.920** | 0.920 |
| | dress-sales | **0.598** | 0.590 | **0.593** | 0.593 |
| | horse-colic | 0.790 | **0.794** | 0.784 | **0.784** |
| | mammographic-masses | 0.865 | **0.866** | **0.867** | 0.867 |
| | nomao | **0.986** | 0.984 | **0.988** | 0.988 |
| | primary-tumor | **0.769** | 0.769 | **0.769** | 0.769 |
| | soybean | 0.999 | **0.999** | **0.999** | 0.999 |
| | thyroid0387 | 0.964 | **0.965** | 0.964 | **0.964** |
| SVM-G | adult | 0.895 | **0.896** | **0.897** | 0.897 |
| | agaricus-lepiota | **1.000** | **1.000** | **1.000** | **1.000** |
| | bands | 0.855 | **0.856** | 0.865 | **0.867** |
| | crx | **0.926** | 0.925 | **0.927** | 0.927 |
| | dress-sales | **0.618** | 0.609 | **0.620** | 0.614 |
| | horse-colic | 0.768 | **0.774** | 0.771 | **0.774** |
| | mammographic-masses | 0.840 | **0.843** | **0.845** | 0.843 |
| | nomao | **0.991** | 0.991 | 0.992 | **0.992** |
| | primary-tumor | 0.762 | **0.764** | 0.765 | **0.766** |
| | soybean | 0.999 | **0.999** | 0.999 | **0.999** |

Table 9: AUROC, additional experiment for imputation of categorical attributes (mode imputation or mean imputation after one-hot encoding). **Bold**: higher value.

| Classifier | Dataset | Without missing-indicators | | With missing-indicators | |
|---|---|---|---|---|---|
| | | Mode | Mean | Mode | Mean |
| LR | thyroid0387 | **0.978** | 0.978 | **0.978** | 0.978 |
| | adult | 0.905 | **0.906** | 0.906 | **0.906** |
| | agaricus-lepiota | **1.000** | **1.000** | **1.000** | **1.000** |
| | bands | **0.819** | 0.814 | **0.833** | 0.832 |
| | crx | **0.924** | 0.924 | 0.923 | **0.924** |
| | dress-sales | **0.620** | 0.611 | 0.620 | **0.620** |
| | horse-colic | **0.789** | 0.788 | 0.786 | **0.787** |
| | mammographic-masses | 0.866 | **0.867** | 0.868 | **0.868** |
| | nomao | **0.986** | 0.984 | 0.988 | **0.988** |
| | primary-tumor | 0.773 | **0.773** | 0.776 | **0.776** |
| | soybean | 0.999 | **0.999** | 0.999 | **0.999** |
| MLP | thyroid0387 | **0.974** | 0.974 | 0.975 | **0.975** |
| | adult | 0.890 | **0.891** | **0.890** | 0.890 |
| | agaricus-lepiota | **1.000** | **1.000** | **1.000** | **1.000** |
| | bands | 0.871 | **0.874** | 0.879 | **0.882** |
| | crx | 0.902 | **0.902** | **0.906** | 0.906 |
| | dress-sales | **0.549** | 0.540 | **0.553** | 0.549 |
| | horse-colic | 0.714 | **0.727** | 0.744 | **0.749** |
| | mammographic-masses | **0.845** | 0.844 | 0.840 | **0.841** |
| | nomao | **0.991** | 0.991 | **0.991** | 0.991 |
| | primary-tumor | 0.768 | **0.769** | **0.782** | 0.781 |
| | soybean | **0.999** | 0.999 | **0.999** | 0.999 |
| CART | thyroid0387 | 0.988 | **0.989** | 0.988 | **0.988** |
| | adult | **0.844** | **0.844** | **0.844** | **0.844** |
| | agaricus-lepiota | **0.991** | **0.991** | **0.992** | 0.991 |
| | bands | **0.749** | 0.744 | **0.759** | 0.757 |
| | crx | 0.897 | **0.899** | 0.897 | **0.899** |
| | dress-sales | **0.568** | 0.568 | **0.570** | 0.568 |
| | horse-colic | **0.742** | 0.728 | **0.724** | 0.723 |
| | mammographic-masses | **0.823** | 0.822 | **0.823** | 0.821 |
| | nomao | **0.916** | **0.916** | **0.916** | **0.916** |
| | primary-tumor | 0.703 | **0.739** | 0.707 | **0.738** |
| | soybean | 0.990 | **0.995** | 0.991 | **0.995** |
| RF | thyroid0387 | **0.909** | **0.909** | **0.909** | **0.909** |
| | adult | 0.890 | **0.891** | **0.890** | 0.890 |
| | agaricus-lepiota | **1.000** | **1.000** | **1.000** | **1.000** |
| | bands | 0.893 | **0.895** | **0.896** | 0.890 |
| | crx | 0.932 | **0.933** | **0.931** | 0.930 |
| | dress-sales | **0.591** | 0.589 | **0.606** | 0.589 |
| | horse-colic | 0.800 | **0.802** | 0.791 | **0.795** |
| | mammographic-masses | 0.812 | **0.823** | 0.821 | **0.822** |
| | nomao | **0.994** | 0.994 | **0.994** | 0.994 |
| | primary-tumor | 0.749 | **0.753** | 0.758 | **0.759** |
| | soybean | 0.999 | **0.999** | 0.999 | **0.999** |
| ERT | thyroid0387 | **0.994** | 0.994 | **0.994** | 0.993 |
| | adult | 0.847 | **0.848** | **0.847** | 0.847 |
| | agaricus-lepiota | **1.000** | **1.000** | **1.000** | **1.000** |

Table 9: AUROC, additional experiment for imputation of categorical attributes (mode imputation or mean imputation after one-hot encoding). **Bold**: higher value.

| Classifier | Dataset | Without missing-indicators | | With missing-indicators | |
|---|---|---|---|---|---|
| | | Mode | Mean | Mode | Mean |
| | bands | 0.890 | **0.893** | **0.890** | 0.889 |
| | crx | **0.914** | 0.914 | **0.914** | 0.914 |
| | dress-sales | 0.572 | **0.589** | **0.602** | 0.591 |
| | horse-colic | 0.799 | **0.806** | 0.782 | **0.785** |
| | mammographic-masses | 0.795 | **0.804** | **0.802** | 0.801 |
| | nomao | **0.994** | 0.994 | **0.994** | 0.994 |
| | primary-tumor | 0.705 | **0.711** | **0.714** | 0.713 |
| | soybean | 0.999 | **0.999** | 0.999 | **0.999** |
| | thyroid0387 | **0.987** | 0.987 | **0.987** | 0.987 |
| ABT | adult | 0.915 | **0.915** | 0.915 | **0.915** |
| | agaricus-lepiota | **1.000** | **1.000** | **1.000** | **1.000** |
| | bands | 0.806 | **0.806** | **0.806** | 0.805 |
| | crx | 0.905 | **0.906** | **0.906** | 0.904 |
| | dress-sales | **0.590** | 0.582 | **0.582** | 0.579 |
| | horse-colic | 0.753 | **0.763** | 0.752 | **0.764** |
| | mammographic-masses | 0.856 | **0.857** | 0.857 | **0.858** |
| | nomao | **0.987** | **0.987** | **0.987** | **0.987** |
| | primary-tumor | **0.661** | 0.640 | **0.660** | 0.639 |
| | soybean | **0.863** | 0.859 | 0.871 | **0.873** |
| | thyroid0387 | **0.685** | **0.685** | **0.685** | **0.685** |
| GBM | adult | 0.927 | **0.927** | **0.927** | 0.927 |
| | agaricus-lepiota | **1.000** | **1.000** | **1.000** | **1.000** |
| | bands | **0.855** | 0.855 | **0.857** | 0.854 |
| | crx | 0.934 | **0.934** | 0.933 | **0.934** |
| | dress-sales | **0.608** | 0.606 | **0.614** | 0.608 |
| | horse-colic | 0.789 | **0.792** | 0.783 | **0.788** |
| | mammographic-masses | **0.857** | 0.857 | **0.859** | 0.858 |
| | nomao | **0.994** | 0.994 | 0.994 | **0.994** |
| | primary-tumor | 0.766 | **0.770** | 0.767 | **0.769** |
| | soybean | 0.999 | **0.999** | 0.999 | **0.999** |
| | thyroid0387 | 0.913 | **0.914** | **0.923** | 0.923 |

