# OpenReview forum: "No imputation without representation"
_TMLR — Rejected by TMLR_

### Review · Reviewer_ooBi · 2022-08-25

**Summary Of Contributions:**

The authors do not propose a new method, but aim to show the superiority of a _strategy_: including missingness indicators or not when dealing with datasets that contain missingness. I find this an interesting take on a paper and welcome it very much. The concluding the authors draw from their experiments is that including a missingness indicator into the input of their models does seem to increase performance, confirming what people generally claim in literature.

**Broader Impact Concerns:**

n.a.

**Requested Changes:**

I have made suggestions with my remarks above.

**Strengths And Weaknesses:**

In general I like the setup of this paper. Much has been said about missingness indicators (informative missingness) in literature, but rarely has it been thoroughly confirmed/disproven. For this, I appreciate the authors' efforts. While I don't see any issues in setup, or focus of this paper, I _do_ have some questions concerning some of the experiments and the interpretation thereof.

* Only reportig p-values is a little opaque, could you pair the p-value with the actual AUROC values? The authors provide quite a few tables, but there is still plenty of room to also add the actual experimental values. Coming from a machine learning background, I am more used to this, and I believe the TMLR audience will also. In short, I believe having both p-values as well as actual AUROC (and perhaps MSE) would give a much clearer picture of the result.



* The authors claim little difference between mean-imputation, mode, iterative, etc. However, to me it seems that replacing a missing value with a constant (such as mean or mode), would likely _leak_ missingness indicators into the input. Wouldn't an adequate model pick up on this one fixed value quite easily? As a dummy experiment, would it make sense to "impute" a missing value with either a complete non-sensical value (such as 250 for the attribute "age"), or a random value and see how that changes performance?

* Right under the section 4.1 title, the authors state that the CART performance is notably worse than others. However, judging from the provided tables, this is not clear to me. Perhaps my first suggestion would shed some more light onto this?

* I'm a bit confused how the "CART overfits" hypothesis is tested. You found it may overfit when using missingness indicators, but it seems you included a pruning technique without missingness (i.e. imputed?) samples? I'm referring here to the second paragraph under Fig 1. Could you clear this up?

* To me it is not clear how Fig 1 relates to missingness. Am I correct in assuming that this figure tells us that some models (such as GMB) may overfit on certain datasets when not stopped in time? This seems like a very general, and already known/accepted, statement. I would suggest to give some more explanation in this figure caption as well as discuss more in the text. If it is indeed there to show that early stopping prevents overfitting, I would suggest removing it from the paper and instead cite Prechelt's seminal paper (https://link.springer.com/chapter/10.1007/3-540-49430-8_3).

* While I truly appreciate the effort of testing on so many datasets, I believe that– especially for the threshold experiments –testing on synthetic data yields a clearer picture. It could be (likely) that removing attributes is having a massive influence on predictive performance, not the amount of missingness.

Beyond comments about the experiments, there are some more general comments.

* The authors claim that including missingness indicators into a model is usually done by first imputing and then completing the sample with a binary vector which encodes missingness into the input. This is fair and probably the go-to method for including missingness into a model. However, there are definitely other ways of doing this also. For example, xgboost can deal with `np.nan` values out of the box (such as xgboost https://xgboost.readthedocs.io/en/stable/faq.html#how-to-deal-with-missing-values or Quinlan's C4.5 algorithm https://en.wikipedia.org/wiki/C4.5_algorithm#Improvements_from_ID.3_algorithm). Typical to these additional algorithms is adjusting the impurity function to split based on a missing value also.

* It would be helpful to state upfront in the experiments section what you wish to show. I found it hard to follow at times as the text is basically running through the completed experiments one-by-one without a clear point that the authors wish to show.

---

> ### Author Response · Authors · 2022-09-14
> **Answers to comments**
>
> >Only reportig p-values is a little opaque, could you pair the p-value with the actual AUROC values?
>
> We have now added an appendix with three additional tables with the actual AUROC scores. In addition, the code and raw results are provided as supplementary material.
>
> >The authors claim little difference between mean-imputation, mode, iterative, etc. However, to me it seems that replacing a missing value with a constant (such as mean or mode), would likely leak missingness indicators into the input. Wouldn't an adequate model pick up on this one fixed value quite easily?
>
> We discuss this in Subsection 4.3, where we argue and provide some experimental evidence that mean imputation leaks more information than mode imputation (because the mean is generally not part of the non-missing values, while the mode by definition is).
>
> We have now also added a paragraph to the introduction, to frame this as a potential theoretical argument against missing-indicators being necessary. Thank you for stressing the importance of this point.
>
> >Right under the section 4.1 title, the authors state that the CART performance is notably worse than others. However, judging from the provided tables, this is not clear to me. Perhaps my first suggestion would shed some more light onto this?
>
> In Table 3, CART has p-values of 0.84, 0.75 and 0.70, which is much higher than the other classifiers.
>
> >I'm a bit confused how the "CART overfits" hypothesis is tested. You found it may overfit when using missingness indicators, but it seems you included a pruning technique without missingness (i.e. imputed?) samples? I'm referring here to the second paragraph under Fig 1. Could you clear this up?
>
> If you are referring specifically to "we repeat our experiment for CART and mean imputation", we mean that we only repeat our experiment with pruning included for mean imputation, not for nearest neighbour or iterative imputation. We obtain new results both without and with missing-indicators, find that now (with pruning) with missing-indicators performs better.
>
> >To me it is not clear how Fig 1 relates to missingness. Am I correct in assuming that this figure tells us that some models (such as GMB) may overfit on certain datasets when not stopped in time? This seems like a very general, and already known/accepted, statement.
>
> We discuss this figure in the last paragraph of Subsection 4.1. The figure illustrates that for GBM, the default number of iterations (100) can lead to both under- and overfitting, depending on the dataset. You are of course correct that this is a general phenomenon, but the documentation of scikit-learn explicitly claims that "Gradient boosting is fairly robust to over-fitting, so a large number usually results in better performance." The purpose of this figure is to illustrate that this is not true for a lot of our datasets, and that even the default of 100 leads to overfitting in some, while at the same time not being high enough for other datasets.
>
> >While I truly appreciate the effort of testing on so many datasets, I believe that– especially for the threshold experiments –testing on synthetic data yields a clearer picture. It could be (likely) that removing attributes is having a massive influence on predictive performance, not the amount of missingness.
>
> For the threshold experiments, removing attributes is not a factor in the performance difference between with and without indicators, because we remove the same attributes in both cases (it is a like-for-like comparison).
>
> It is true that testing on synthetic data yields a clearer picture, but we feel that this has already received attention in the literature. It is precisely because the situation for real-life data is less clear that we wanted to perform this experiment on real-life datasets.
>
> >The authors claim that including missingness indicators into a model is usually done by first imputing and then completing the sample with a binary vector which encodes missingness into the input. This is fair and probably the go-to method for including missingness into a model. However, there are definitely other ways of doing this also. For example, xgboost can deal with np.nan values out of the box [...] or Quinlan's C4.5 algorithm [...].
>
> Whenever the implementation that a practitioner is using has a specific way to handle missing values, then that should be used. We have tried to make this clearer in the introduction, and have listed a number of references to such proposals in Subsection 3.4.
>
> >It would be helpful to state upfront in the experiments section what you wish to show. I found it hard to follow at times as the text is basically running through the completed experiments one-by-one without a clear point that the authors wish to show.
>
> We have inserted Subsection 3.1 with the three questions that we answer, and renamed the subsection titles of Section 4 to match these questions.

---

### Review · Reviewer_9ecM · 2022-08-27

**Summary Of Contributions:**

This paper provides an empirical evaluation of how three different imputation strategies work in conjunction with a downstream classification task (accuracy is evaluated on this classification task), where the classifier either has or does not have access to missingness indicators for the different variables.

**Requested Changes:**

Please see the list of weaknesses I stated above. I think these points can all be addressed with a revised draft.

**Strengths And Weaknesses:**

Strengths:
- I agree with the author(s) that there by and large has not been a systematic comparison of classification after imputation, comparing different imputation approaches and whether missingness indicators are used.
- The authors consider a relatively large collection of datasets compared to existing work on imputation.

Weaknesses:
- In Section 3.3, it is mentioned that default hyperparameter values are used for the different classification algorithms (except for the exception that is mentioned for SVM-L, LR, and MLP). I would suggest actually trying a wider hyperparameter grid per algorithm because some of the algorithms are quite sensitive to hyperparameters and I do not think the default options sklearn uses are necessarily the "best". In particular, in the current draft of the paper, some algorithms might appear a lot worse than they actually are if you allow for hyperparameter tuning. I also wouldn't be surprised if the best hyperparameter setting varies depending on which imputation strategy is used and also depending on whether missingness indicators are available or not. Because of how important hyperparameter tuning often is in practice (and that practitioners will typically be doing hyperparameter tuning), by not including this, I think that the empirical results of the paper as presented now are suspect.
- I'm not convinced that presenting p-values as the main way to compare methods is the "right" approach. I realize that this is a somewhat of a subjective point, but perhaps presenting something closer to the raw accuracy values would be better. For example, per dataset, and per classification method, perhaps what could be reported is the ratio of the accuracy achieved with/without missingness indicators for a specific imputation strategy vs a single "standard" reference point (a reasonable choice of this standard reference point could be to use mean imputation without missingness indicators). Basically using this information, it would be possible to back out what all the raw accuracy scores are if one wanted to. By reporting p-values instead, it is not straightforward how to back out the raw accuracy values.
- I think the author(s) should explicitly also mention and discuss the MNAR setting that various researchers have now explored (there are lots of papers on this; for instance, see the textbook by van Buuren 2018 or Morvan et al 2021).
- I'll remark that the theory of Morvan et al (2021) that is on regression also applies to classification (see their note at the end of their Section 3 on "Impute-then-classify procedures"). How does their theory (that can extend to classification) relate to what your empirical findings are? Note that these authors have also explored the impact of a having a missingness mask (on page 10 of their paper, one of their main findings is that "Adding the mask is critical in MNAR settings with mean and MICE imputations"). The impression I have is that your experimental findings are largely just confirming what these other authors also find, but with a larger collection of datasets? It would be great if the author(s) could clarify what new generalizable insights have been obtained in the empirical study in this paper compared to known findings by other authors such as Morvan et al.

References:
- Stef van Buuren. Flexible Imputation of Missing Data. CRC Press 2018.
- Marine Le Morvan, Julie Josse, Erwan Scornet, Gaël Varoquaux. What's a good imputation to predict with missing values? NeurIPS 2021.

---

> ### Author Response · Authors · 2022-09-14
> **Answers to comments**
>
> >I would suggest actually trying a wider hyperparameter grid per algorithm because some of the algorithms are quite sensitive to hyperparameters and I do not think the default options sklearn uses are necessarily the "best". In particular, in the current draft of the paper, some algorithms might appear a lot worse than they actually are if you allow for hyperparameter tuning. I also wouldn't be surprised if the best hyperparameter setting varies depending on which imputation strategy is used and also depending on whether missingness indicators are available or not. Because of how important hyperparameter tuning often is in practice (and that practitioners will typically be doing hyperparameter tuning), by not including this, I think that the empirical results of the paper as presented now are suspect.
>
> There are four reasons why we decided to work with default hyperparameter values.
>
> 1) Keeping the runtime of our experiments manageable. In particular iterative imputation and some of the classifiers like SVM have a very long runtime with the larger datasets. Even if we only evaluated two different hyperparameter values with five-fold cross-validation, the runtime would increase about tenfold.
> 2) Implementing good hyperparameter tuning requires a number of non-trivial choices for each classifier: which hyperparameters to tune, which values to try (+ how to parametrise them), and which optimisation algorithm to use, and we did not wish for this to divert from the main focus of this paper.
> 3) Whether to use missing-indicators (and which kind of imputation to use) is part of the model-selection process, so if a user is doing hyperparameter tuning, they could also decide on the use of missing-indicators through validation on the training set. The purpose of our paper is to establish a good default choice.
> 4) We have no reason to believe that any of the default hyperparameter choices in scikit-learn are less optimal without missing-indicators than with.
>
> If there is a particular hyperparameter whose default value in your consideration could be biased in favour of missing-indicators, and you would like us to try an alternative value, we could conduct an additional experiment with mean/mode imputation to test this.
>
> >I'm not convinced that presenting p-values as the main way to compare methods is the "right" approach. I realize that this is a somewhat of a subjective point, but perhaps presenting something closer to the raw accuracy values would be better.
>
> We have now added an appendix with three additional tables with actual AUROC scores (as also requested by one of the other reviewers).  In addition, the code and raw results are provided as supplementary material.
>
> >I think the author(s) should explicitly also mention and discuss the MNAR setting that various researchers have now explored (there are lots of papers on this; for instance, see the textbook by van Buuren 2018 or Morvan et al 2021).
>
> We now mention MNAR explicitly in the introduction, to make it clear that we are talking about MNAR when referring to informative missing values.
>
> >I'll remark that the theory of Morvan et al (2021) that is on regression also applies to classification (see their note at the end of their Section 3 on "Impute-then-classify procedures"). How does their theory (that can extend to classification) relate to what your empirical findings are? Note that these authors have also explored the impact of a having a missingness mask (on page 10 of their paper, one of their main findings is that "Adding the mask is critical in MNAR settings with mean and MICE imputations"). The impression I have is that your experimental findings are largely just confirming what these other authors also find, but with a larger collection of datasets? It would be great if the author(s) could clarify what new generalizable insights have been obtained in the empirical study in this paper compared to known findings by other authors such as Morvan et al.
>
> Thank you for pointing Le Morvan et al 2021 out to us. We have included it in our overview of previous experiments, and have added a paragraph to the introduction about the imputation manifolds that they identify.
>
> The largest difference is not just that we use more datasets, but that Le Morvan et al 2021 simulate MCAR and MNAR missing values in artificial datasets (which is great for establishing possible scenarios), whereas we use natural missing values in real-life datasets, to establish whether missing values are sufficiently MNAR in practice for missing-indicators to be beneficial.
>
> The second difference is that they only consider MLP, whereas we evaluate a number of different classifiers, which is relevant for their argument that a sufficiently strong classifier should be able to recover the missingness information from the imputation manifolds, even without missing-indicators.

---

### Review · Reviewer_4w1B · 2022-08-30

**Summary Of Contributions:**

The authors consider the problem of supervised learning with missing values.

A standard and slightly naive approach to this problem is just to impute the data using single imputation (e.g. by the mean, or a more complex algorithm), and then feed the imputed data to a standard supervised algorithm. This is problematic for many reasons, notably because the supervised algorithm may not be able to differentiate between "true" values and imputed ones. A commonly used fix is to concatenate the missingness pattern to these imputed data, and finally to feed this to a standard supervised algorithm. By feeding the missingness pattern to the supervised model, there is no loss of information at all, since the algorithm can use this pattern to identify which values are missing.

The main goal of the paper is to study experimentally the benefits of this fix (called the missing-indicator approach), on many data sets that inherently contain missing values, and using many different imputations and supervised algorithms.

Their experiments show that the missing-indicator approach is very generally useful.

**Broader Impact Concerns:**

I have no particular concern about this work.

**Requested Changes:**


In spite of my lengthy "weaknesses" section, I think that this paper is a nice contribution to the field of missing data. I believe the following modifications would strengthen the paper:

1) Acknowledging the existence of non-imputation-based techniques, and that the missing indicator can be useful for these techniques as well


2) Acknowledging that there exist some versions of the algorithms used by the authors that can handle missing values in a more sensible way that simply using imputation.

3) Add maybe to the empirical comparisons some comparisons with such non-imputation-based techniques. One that would be not too hard to add, since you mostly use sk learn, would be the "missing incorporated in attribute" that is readily implemented in the HistGradientBoostingClassifier function.


**Strengths And Weaknesses:**



Strengths


The paper reads is very nicely written, and I think the empirical study is essentially well done. The general message that the missing-indicator is empirically useful is a nice contribution.

The paper tackles a relevant question, and gives a clear enough empirical answer. As a practitioner of supervised learning with missing values, I think that the good performance of missing-indicator approach is known as folklore, so it is really nice that there exist now a good empirical study to back this folk knowledge.


Weaknesses

I have two (related) major concerns about this paper. Both concern the fact I do not believe that the question of the utility of the missing-indicator is tightly related to  (single) imputation, as the title and the paper suggest. There are many other approaches than single imputation to the problem of supervised learning with missing values, and virtually all of these can benefit from the missing-indicator. The existence of such methods is not really acknowledged of discussed by the authors. More importantly, they include several techniques for which techniques more refined than single imputation are very popular.

1) My first concern is that the paper seems to imply that single imputation (with, or without the missing-indicator) is the standard (and only) approach to supervised learning to missing values. I think it would be quite important to acknowledge that there exist several other approaches that do not directly rely on imputations (yet can also be used with or without the missing indicator). Here are some examples (I do not expect/request that the authors cite all of these papers of course, these are just illustrations that there are many ways to go beyond single imputation for supervised learning):
- multiple imputation has been used a lot in a supervised context for a long time (e.g. Little, 1992, Section 7)
- there is an important literature on linear regression, two recent proposals that use both the mask and some imputations are Sportisse et al. (2020) and Ayme et al. (2022)
- for neural nets, the work of Tresp and co-authors in the 90s (Ahmad and Tresp, 1993, Tresp et al., 1994, 1995) and the closely related technical report of Ghahramani and Jordan (1995) aim at marginalising over the missing values (akin to multiple imputation, in a sense). Recent iterations of this line of work include Smieja et al. (2018) or Ipsen et al. (2022)
- for decision trees, there are specific techniques that handle missing values, and that implicitly use the missingness mask. A popular one is for instance the "missing incorporated in attribute" (MIA) of Twala et al. (2008) that is implemented in both XGboost and scikit-learn. A theoretical and empirical discussion on trees was recently provided by Josse et al. (2020). In particular, they argue that the missing indicator approach is quite efficient.
- there are some neural architectures that have been designed to handle missing values without imputation, for instance Ma et al. (2019) used permutation-invariant neural nets (in an unsupervised context, and they were then used for supervised learning by Ipsen et al., 2022). Danel et al. (2020) used graph neural nets to generalise convolutional nets to incomplete images.


2) The authors use several techniques (CART, RF, GBM) that do not really require to use imputation. For these methods, they still use their usual pipeline of imputing then classifying. I think it would make a lot of sense to consider also the versions that do not use imputation (e.g. MIA) as they are the most popular, and can also be used with, or without missing indicators. In any case, the authors should clarify that the versions of the algorithms they are using are not necessarily the most well suited for handling missing values.

I also have a more minor concern: I do not really agree with the sentence "the usefulness of missing-indicators in machine learning is ultimately an empirical question."  I feel like it is very relevant to tackle the question from an empirical perspective (which is why I enjoyed your paper), but theoretical treatments around this question are also valuable. For instance, the recent work of Le Morvan et al. (2021) shows that, for sufficiently flexible classifiers, the classifier can "learn the mask" via the imputation (because the imputed values live in so called "imputation manifolds" of lower dimension. Such "imputation manifolds" were also empirically noticed by Ipsen et al. (2022).



Additional references

- Ahmad and Tresp. Some solutions to the missing feature problem in vision, NeurIPS 1993
- Ayme et al., Near-optimal rate of consistency for linear models with missing values, ICML 2022
- Ghahramani and Jordan. Learning from incomplete data. Technical report, 1995
- Ipsen et al. How to deal with missing data in supervised deep learning? ICLR 2022
- Josse et al. On the consistency of supervised learning with missing values. arXiv:1902.06931, 2020
- Le Morvan et al., What’s a good imputation to predict with missing values? NeurIPS 2021
- Little. Regression with missing Xs: a review. JASA 1992
- Twala et al. Good methods for coping with missing data in decision trees. Pattern Recogn. Lett., 2008
- Ma et al., EDDI: Efficient dynamic discovery of high-value information with partial VAE. ICML 2019
- Danel et al., Processing of Incomplete Images by (Graph) Convolutional Neural Networks, ICONIP 2020
- Smieja et al. Processing of missing data by neural networks. NeurIPS 2018
- Sportisse et al., Debiasing Stochastic Gradient Descent to handle missing values, NeurIPS 2020
- Tresp et al. Training neural networks with deficient data, NeurIPS 1994
- Tresp et al. Efficient methods for dealing with missing data in supervised learning, NeurIPS 1995

---

> ### Author Response · Authors · 2022-09-14
> **Reply to requested changes**
>
> >I also have a more minor concern: I do not really agree with the sentence "the usefulness of missing-indicators in machine learning is ultimately an empirical question." I feel like it is very relevant to tackle the question from an empirical perspective (which is why I enjoyed your paper), but theoretical treatments around this question are also valuable.
>
> We have now tweaked this sentence to say "In light of these conflicting theoretical arguments, the usefulness of missing-indicators for real-life machine learning problems is an interesting empirical question."
>
> >For instance, the recent work of Le Morvan et al. (2021) shows that, for sufficiently flexible classifiers, the classifier can "learn the mask" via the imputation (because the imputed values live in so called "imputation manifolds" of lower dimension. Such "imputation manifolds" were also empirically noticed by Ipsen et al. (2022).
>
> Thank you for pointing out Le Morvan et al. (2021). We have added a paragraph to the introduction to discuss this phenomenon. Incidentally, it corresponds with what we suggest in Subsection 4.3, that mean imputation allows some missingness-information to be recovered, because the mean (unlike the mode) is typically not one of the non-missing values.
>
>
> >In spite of my lengthy "weaknesses" section, I think that this paper is a nice contribution to the field of missing data. I believe the following modifications would strengthen the paper:
>
> >Acknowledging the existence of non-imputation-based techniques, and that the missing indicator can be useful for these techniques as well
>
> >Acknowledging that there exist some versions of the algorithms used by the authors that can handle missing values in a more sensible way that simply using imputation.
>
> We have tried to make this more explicit in the introduction, added a paragraph to Subsection 3.4 (where we list the classifiers) with a number of references to classifier-specific proposals (kindly provided by you), and added a suggestion in the Conclusion that missing-indicators may also be combined with classifier-specific proposals.
>
> We are in particular grateful for your pointing out Twala et al 2008, and we have added a paragraph to Subsection 2.2 (Previous experiments) that MIA can be understood as implicitly using missing-indicators, and has been shown to outperform imputation.
>
> >Add maybe to the empirical comparisons some comparisons with such non-imputation-based techniques. One that would be not too hard to add, since you mostly use sk learn, would be the "missing incorporated in attribute" that is readily implemented in the HistGradientBoostingClassifier function.
>
> While we agree that missing-indicators could also be useful in combination with some of the non-imputation based classifier-specific approaches, it seems to us that missing-indicators don't have anything to offer in addition to MIA in particular, since that already allows splitting on missingness (as you point out above). (We have repeated our experiment on the HistGradientBoostingClassifier and find that adding missing-indicators doesn't affect performance much (p=0.66).) So we are not sure that it would be very useful to add this to the paper.

---

### Author Response · Authors · 2022-09-14
**New draft**

We thank all three reviewers for their appreciation of our paper, and the very useful comments. We have uploaded a new draft of our paper.

---

### Decision · Action_Editors · 2022-10-09

**Recommendation:** Reject

**Comment:**

My main concern is that the provided experimentation is not enough to provide a clear and actionable recommendation to practitioners.

First, results do not strongly support the claim. As highlighted by reviewers during the discussion, reporting only p-values (e.g., in Tables 3,4) is not enough to evaluate the impact of including missing indicators. The additional results reported by the authors in the appendix (Tables 7-9) do not bring enough evidence for the made claim that missing indicators help: most of the times the values for the two different hypothesis are very close (e.g., one point on the third significant digit) or exactly the same, despite being highlighted in bold (if results are rounded, all values should be highlighted in bold). This is even more crucial for the UCI datasets with small test samples.
This suggests that what is being observed might be noise (despite being significant, once aggregated in Tables 3,4 under a Wilcoxon test).

Second, the methods involved are employed by using the default hyperparameters set of scikit-learn. As one reviewer highlighted, this might introduce a latent factor biasing the results. I agree with the reviewer that running a systematic hyperparameter search would help control for this latent factor. This is highlighted, for instance, by using or not pruning with CART. One of the few hyperparameters the authors switch on and off.
However, one shall not pick the best values from such grid search. Instead, the distribution across the different parameter values should be used to decide if there is a systematic improvement across hyperparameters due to using missing indicators.

Concerning the consequent computational effort needed, using a random grid search could help alleviate the issue. However, to strongly support claims, sometimes investing time and effort in strong experimental design is necessary.

Furthermore, these results are not discussed enough in the main body of the paper. I appreciated that authors augmented their background sections and discussed more other (theoretical and not) literature on missing indicators and values. What is still missing is a deeper understanding why the results would support a claim. For example, Tables 3 and 4 is commented in a few lines and Sec 4.1 ultimately is only a discussion on why two methods are not behaving according to the claim.

**Audience:**

The scope of the paper is relevant to many machine learning and data science practitioners, who have to deal with missing values on a daily basis and are possibly using an out-of-the-box implementation of classical learning algorithms.

Extensive empirical investigations are much needed in a time when the machine learning model zoo is growing out of control. Conducting them properly, however, requires special care, and time.

**Claims And Evidence:**

The paper proposes an empirical investigation to assess what is the impact of encoding missing values entries with indicators in addition to imputing missing values. They do so by evaluating classical machine learning models as implemented in scikit-learn (variants of support-vector machines, decision trees, nearest neighbor and multi-layer perceptrons) on a number of UCI datasets and missing value imputation techniques (mean and median).

The main claim is that including missing indicators is generally beneficial for downstream classification. To this end, authors provided a number of statistical tests when comparing the classifier performances with and without missing indicators. However, even if the p-values can be somehow ``significative'', the raw performances (reported as rebuttal) differ by little. It then become a question if the observed value differences are just an effect of noise. Running a systematic hyperparameter grid search, would help establish this. See comments below.

---

> ### Author Response · Authors · 2022-10-12
> **Request for clarification**
>
> Thank you for your additional comments. We are of course disappointed: we felt that all three reviews were essentially positive and that we had addressed their requested changes.
>
> We are aware that you have made your decision, but it would be of great help to us if you could clarify a few of the points that you raise.
>
> >My main concern is that the provided experimentation is not enough to provide a clear and actionable recommendation to practitioners.
>
> >most of the times the values for the two different hypothesis are very close (e.g., one point on the third significant digit) or exactly the same
>
> We agree that the effect size is not very large. We would have been happy to point this out in the paper, and amend our conclusion accordingly. (If, as you say, the question that we try to answer is relevant to many people, then even a negative result (that missing-indicators do not increase performance), would be interesting.)
>
> >This is even more crucial for the UCI datasets with small test samples. This suggests that what is being observed might be noise (despite being significant, once aggregated in Tables 3,4 under a Wilcoxon test).
>
> It is precisely to reduce the effect of random variation that we have repeated all our experiments for 5 different random states (aggregating the resulting AUROC scores). If the result were due to noise, we wouldn't expect the noise to affect the results all in the same direction. The p-values quantify the chance that the results are coincidence, due to the selection of datasets.
>
> >Second, the methods involved are employed by using the default hyperparameters set of scikit-learn. As one reviewer highlighted, this might introduce a latent factor biasing the results. I agree with the reviewer that running a systematic hyperparameter search would help control for this latent factor.
>
> We chose to use default hyperparameter values because as you point out above, practitioners "are possibly using an out-of-the-box implementation of classical learning algorithms". If users apply hyperparameter optimisation, they could include the use of missing-indicators as an additional hyperparameter.
>
> However, as we indicated to the reviewer, we wouldn't mind performing additional experiments, but this would necessarily require a selection of additional hyperparameter values to test, so we would like to do this in a targeted way. Could you point out specific hyperparameters whose default values you are concerned about might be biased in favour of missing-indicators?
>
> >This is highlighted, for instance, by using or not pruning with CART. One of the few hyperparameters the authors switch on and off. However, one shall not pick the best values from such grid search. Instead, the distribution across the different parameter values should be used to decide if there is a systematic improvement across hyperparameters due to using missing indicators.
>
> We only looked at pruning as an additional experiment, because there was a clear mechanism that might bias the result with the default hyperparameter values against missing-indicators: overfitting. We pointed out that even without missing-indicators, pruning increased performance for CART. So we would disagree with your characterisation that we "picked the best value from such grid search"; we reported both results, and explicitly stated in our conclusion that missing-indicators decreased performance for CART. If you feel that this could be made clearer in the text, we would be happy to receive suggestions.
>
> >Furthermore, these results are not discussed enough in the main body of the paper. I appreciated that authors augmented their background sections and discussed more other (theoretical and not) literature on missing indicators and values. What is still missing is a deeper understanding why the results would support a claim. For example, Tables 3 and 4 is commented in a few lines and Sec 4.1 ultimately is only a discussion on why two methods are not behaving according to the claim.
>
> Simply put, we already know why missing-indicators should or should not improve performance, as discussed in the introduction. So the question that we try to answer is a very simple empirical one: in real-life datasets, are missing values sufficiently informative and can classifiers make sufficiently use of this information that missing-indicators increase performance?
>
> In addition, Section 4.2 provides a further analysis of when precisely missing-indicators start to become useful in terms of missingness.

---

> > ### Comment · Action_Editors · 2022-10-14
> > **Further clarification**
> >
> > Dear authors, thanks for your feedback.
> >
> > I based my decision on the discussion as well as on the reviewers' recommendations: only one (out of three) was leaning towards acceptance.
> > I hope the following comments are addressing your remaining concerns.
> >
> > > We agree that the effect size is not very large. We would have been happy to point this out in the paper, and amend our conclusion accordingly.
> >
> > I agree that negative results are very valuable. I believe that changing your claim at this point (from "missing indicators definitely help" to "missing indicators might not help") will go beyond a minor revision. You are welcome to resubmit a refactored version of this work, if you do not disclose your identities.
> >
> > > for 5 different random states (aggregating the resulting AUROC scores). If the result were due to noise, we wouldn't expect the noise to affect the results all in the same direction.
> >
> > I am referring to a statistical artifact that might not impact practioneers. When on most datasets the AUCROC, when using or not missing indicators, differs by less than 0.001 (or 0.1%), it does not seem impactful and should not be highlight it in bold in Tables nor claimed to be better.
> >
> > > We chose to use default hyperparameter values because as you point out above, practitioners "are possibly using an out-of-the-box implementation of classical learning algorithms".
> >
> > Using an implementation out-of-the-box is generally followed by performing an hyperparameter optimization step. The effect of the hyperparameters might impact overfitting on a specific dataset (as you show for pruning decision trees). See next answer.
> >
> > > However, as we indicated to the reviewer, we wouldn't mind performing additional experiments, but this would necessarily require a selection of additional hyperparameter values to test, so we would like to do this in a targeted way. Could you point out specific hyperparameters whose default values you are concerned about might be biased in favour of missing-indicators?
> >
> > > We only looked at pruning as an additional experiment, because there was a clear mechanism that might bias the result with the default hyperparameter values against missing-indicators: overfitting.
> >
> > Many hyperparameters help alleviate overfitting for the many models you use, and need to be taken into consideration. For example, for logistic regression one might use a weight norm (L1, L2, or combinations thereof) to regularize the model. For k-nearest neighbors, the choice of k and for SVMs the margin. For MLPs one can again use weight regularization and early stopping.
> >
> > > So we would disagree with your characterisation that we "picked the best value from such grid search";
> >
> > I was referring to investigate the effect of missing indicators across the whole distribution of hyperparameter values. I.e., see if the effect of missing indicators is more sensible to certain hyperparameter configurations than others. This is in opposition of doing a grid search and perform the analysis only on the hyperparameter values that minimize some held-out error.

---

> > > ### Author Response · Authors · 2022-10-14
> > > **Thank you**
> > >
> > > Thank you very much for taking the time to respond to our questions.
> > >
> > > >I based my decision on the discussion as well as on the reviewers' recommendations: only one (out of three) was leaning towards acceptance.
> > >
> > > In that case, we regret the fact that the reviewers did not reply to our comments to explain their remaining concerns. We had submitted our replies in the hope that there would be a discussion. In particular because we did very much appreciate their reviews and are grateful that we could improve our manuscript with their help.
> > >
> > > >Many hyperparameters help alleviate overfitting for the many models you use, and need to be taken into consideration. For example, for logistic regression one might use a weight norm (L1, L2, or combinations thereof) to regularize the model. For k-nearest neighbors, the choice of k and for SVMs the margin. For MLPs one can again use weight regularization and early stopping.
> > >
> > > We are still not quite clear how with the hyperparameters that you mention the default values might have been biased in favour of missing-indicators. If the default configuration overfits, then this is detrimental for missing-indicators because they create a higher-dimensional representation of the same data. If, instead, some non-default configuration leads to overfitting, then yes, missing-indicators will probably do worse with that, but that amounts to saying that a classifier can be misconfigured.
> > >
> > > (Note that the scikit-learn implementations of LR and MLP use weight regularization and early stopping by default. )